# Codon-specific Ramachandran plots show amino acid backbone conformation depends on identity of the translated codon

Aviv A. Rosenberg [1,2], Ailie Marx[1,2] & Alex M. Bronstein [1✉]

Synonymous codons translate into chemically identical amino acids. Once considered inconsequential to the formation of the protein product, there is evidence to suggest that codon usage affects co-translational protein folding and the final structure of the expressed protein. Here we develop a method for computing and comparing codon-specific Ramachandran plots and demonstrate that the backbone dihedral angle distributions of some synonymous codons are distinguishable with statistical significance for some secondary structures. This shows that there exists a dependence between codon identity and backbone torsion of the translated amino acid. Although these findings cannot pinpoint the causal direction of this dependence, we discuss the vast biological implications should coding be shown to directly shape protein conformation and demonstrate the usefulness of this method as a tool for probing associations between codon usage and protein structure. Finally, we urge for the inclusion of exact genetic information into structural databases.

[1] Computer Science, Technion – Israel Institute of Technology, Haifa 3200003, Israel. [2] These authors contributed equally: Aviv A. Rosenberg, Ailie Marx. ✉email: alexbronst@gmail.com

One of the most critical cellular processes is the decoding of genetic information into functional proteins. Transfer RNA (tRNA) molecules recognize codons of the messenger RNA (mRNA) sequence as it passes through the ribosome and deliver specific amino acids sequentially for addition to the growing peptide chain. 61 codons map to 20 amino acids, meaning that most amino acids are encoded by more than one, synonymous, codon. Once considered a silent redundancy of the genetic code, synonymous coding is now known to be functionally important, subject to evolutionary selective pressure and clearly associated with disease[1–5]. Changes in synonymous coding can alter mRNA splicing, mRNA folding, and stability[6–8], and can affect translational speed and accuracy and the conformation of the translated protein[9–12].

Numerous studies have shown that changes in the rhythm of translation can alter the kinetics of co-translational folding and so, the global conformation of the final protein product[13,14]. Translation rate is affected by synonymous codon usage which alters mRNA structure and tRNA abundance, the latter coevolving with codon bias[15–20]. This mechanism provides an indirect association between codon usage and global protein structure. Nevertheless, whether and how synonymous variants of a gene will alter the conformation of the final folded protein is still poorly predictable and additionally the literature is riddled with reports of single synonymous mutations causing measurable functional effects that are not well-explained by current mechanisms[21–24]. Together this suggests that we are far from fully understanding the role of codon usage in orchestrating protein folding.

To the best of our knowledge, no studies have investigated whether the specific backbone torsion of an amino acid is associated with the synonymous codon from which it was translated. To probe for such a direct and local association, we developed a method for estimating and comparing codon-specific backbone dihedral angle distributions, which we term codon-specific Ramachandran plots. Comparing these distributions for pairs of synonymous codons, statistically significant differences are observed. Our results demonstrate that the backbone dihedral angle of an amino acid is statistically dependent on the identity of the codon from which it was translated, however, these results cannot shed any light on the causal direction of this dependence.

## Results

### Data collection, codon assignment and development of analysis tools.
The first challenge in investigating the dependence between codon identity and the protein backbone structure is, regrettably, the absence of annotation within the Protein Data Bank (PDB) for the actual genetic template used in producing the protein for crystallization. Automatic assignment of codon identity to each position in a protein structure is a prerequisite to calculate codon-specific Ramachandran plots. It is imperative to stress that any method used for large-scale codon reassignment will carry an inherent limitation of being contaminated with uncertainty and error. The main reason is that codon optimization is very commonly used to improve heterologous gene expression[25], especially in structure determination which necessitates the production of large amounts of soluble protein. There is not one common approach to codon optimization, and the choice of method often depends on trial and error[26,27]. To limit codon assignment errors from including codon-optimized genes, we selected only structures of *E. coli* proteins expressed in *E. coli*, the most common expression system in the PDB. We purposely did not include all natively expressed proteins from other species, since codon biases differ between organisms[28] and such generalization could obfuscate the sought for associations between coding and structure.

The procedure for computing and comparing codon-specific backbone dihedral angle distributions is displayed in Fig. 1 and detailed in the Methods. Briefly, high-resolution PDB structures are retrieved, structures are filtered to remove homology bias and the resulting proteins are grouped according to their unique Uniprot entry. For each position in a protein chain, the backbone dihedral angles, φ and ψ, are calculated; if multiple PDB structures are available, the angles are averaged. Alongside this precise structural information, DSSP secondary structure is designated, and codons are assigned according to the genetic sequences obtained from ENA records cross-referenced in the Uniprot entry. Only locations with unambiguously assigned secondary structures and codons are retained.

We used only well-fitted X-ray crystal structures having a resolution no worse than 1.8 Å (Supplementary Fig. 1), as a recent study considering alternate backbone conformations found resolutions better than 2.0 Å useful for such purposes[29].

A second challenge in investigating the dependence between codon identity and the protein backbone structure is that synonymous codons vary greatly in relative abundance (Supplementary Table 1). The challenge is that any difference we see in terms of the measured distance between estimated distributions, could be due to chance, arising from the availability of only finite data. Thus, our approach is to determine whether the distance we measure is large enough such that the probability of obtaining such a distance by chance from identical underlying distributions is extremely small.

Specifically, our analysis carefully accounts for this by comparing non-parametric distribution estimates which are calculated using the same sample size for all codons in a synonymous group; that of the rarest codon. We combine this with bootstrap-resampling to account for all available data from the abundant codons. We do not assume any specific parametric form of the underlying (i.e., real) distribution of codon dihedral angles because these distributions are complex, unknown, and unlikely to be accurately approximated by any closed-form parametric model. Instead, we aim only to compare the estimated distributions and as such do not require that samples from each codon distribution exhaustively represent the entire underlying distribution. Rather we developed tools and employed existing statistical methods which are sensitive and capable of comparing between nonparametric estimated distributions. We quantify the differences using a distribution-free statistical test to calculate the *p*-value of observing these data under the assumption that the two codons in question have the same underlying distribution. The Methods section describes these details in full and shows additional experiments on synthetic and real data which validate our approach on various sample sizes.

### Codon-specific backbone angle distributions are significantly distinct within the β mode.
Synonymous codons are known to have distinct propensities to different secondary structures[30–33], which is manifested as different probabilities of the corresponding modes in the full codon-specific Ramachandran plots (Fig. 2). The difference in propensity for the main two, α and β, secondary structure modes might therefore dominate the difference between the codon-specific Ramachandran plots of synonymous codons. To factor out this effect, we conditioned the dihedral angle distribution on the secondary structure, effectively restricting it either to the distinct β or α modes. Select examples of the resulting codon-specific Ramachandran plots, conditioned on these modes, are shown (Fig. 3), while the full set is provided (Supplementary Figs. 2, 3). Visually, it is evident that synonymous codons of some amino acids have clearly distinguishable distribution shapes especially in the β-mode.

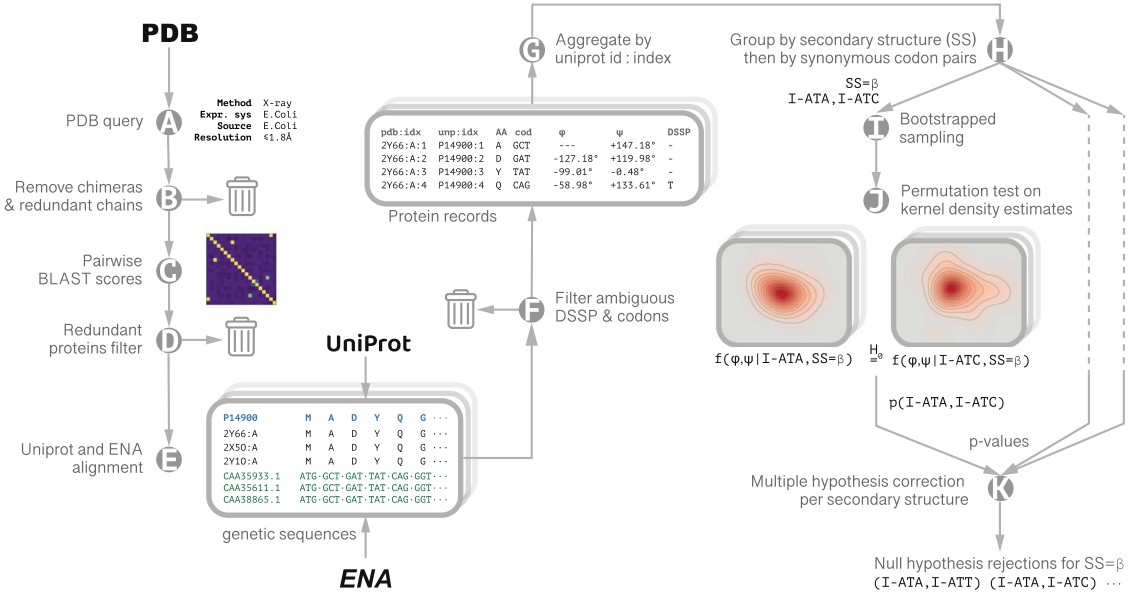

**Fig. 1 Data Collection and Analysis.** Querying the PDB for high resolution (≤1.8 Å), high quality ($R_{free}$ ≤ 24%) X-ray crystal structures of *E. coli* proteins expressed in *E. coli* (**A**), out of which unique chains were extracted (**B**). To ensure the protein set was non-redundant, pairwise sequence alignment scores were calculated between every pair of unique sequences (**C**). A farthest point sampling procedure was then employed to produce a sub-set of structures with normalized pairwise similarity not exceeding 0.7 (**D**). Structures were then grouped according to their unique Uniprot identifier. Genetic sequences were retrieved from ENA records cross-referenced by Uniprot (**E**), adopting a conservative approach: locations having more than one genetic variant for a specific residue are excluded from further analysis (**F**). For each group, a single protein record was generated with each point in the amino acid sequence annotated with the φ, ψ backbone dihedral angles averaged over all the structures in the record, the codon, and DSSP secondary structure assignment (**G**). The final data set included 1343 protein chains. We estimated the codon distributions from their samples using kernel density estimation (KDE) on a torus with a Gaussian kernel width of 2°. We used a bootstrap-resampling scheme to estimate multiple realizations of these codon specific distributions. p-values were calculated via permutation test on the $L_1$ distance between the estimated densities (steps **H**–**J**); the rejection threshold (p = 0.019) was established by Benjamini-Hochberg multiple hypothesis correction with the false discovery rate set to q = 0.05 (**K**).

To quantify those differences and their significance, we used a distribution-free two-sample permutation test with the $L_1$ distance between KDEs serving as the test statistic and assigned p-values to each synonymous codon pair with respect to the null hypothesis that the two codons have the same underlying distribution of backbone angles. To determine the p-value threshold for statistical significance in a setting where multiple hypotheses are considered together, we employed the Benjamini-Hochberg correction with false discovery rate set to 0.05. This process is shown schematically (Fig. 1) and detailed in the Methods. Matrices and multidimensional scaling (MDS) plots visualizing the distances between select pairs of synonymous codon distributions are shown alongside the contour plots (Fig. 3) and for all synonymous codon groups (Supplementary Figs. 4–7).

Note that together with the 87 synonymous pairs, we also included the 61 comparisons of each codon to itself. The latter served as a control, and indeed, the null hypotheses were not rejected for any of the same codon pairs in either of the secondary structures. No synonymous pairs were rejected in comparisons of the distributions of the α-mode, however, when comparing distributions for the β- mode, 57 of the 87 synonymous pairs were rejected (Fig. 4).

It is not surprising that α-helices, being less flexible than β-sheets[34,35], display less variability in codon-specific Ramachandran plots. The Ramachandran plot defines a richer range of structural contexts than the discrete categories available in DSSP annotation[36], especially in the β-mode. It is therefore possible that some of the differences we observe between codon-specific dihedral angle distributions in the β-mode are attributable to codon preferences for finer secondary structure categories such as parallel and antiparallel β-sheets. However, it should be noted that we used a strict conditioning by the secondary structure,

taking only the DSSP annotation[37] E (*extended strand* – β-sheet in parallel and/or anti-parallel sheet conformation with minimum length of 2 residues) for the β-mode, and H (*α-helix* – a 4 turn helix with minimum length of 4 residues), for the α-mode.

Having found synonymous codons which have different dihedral angle distributions within the β-mode, we explored the possibility that synonymous codon preferences for specific positions within this secondary structure (beginning, middle or end; refer to Supplementary Fig. 8)[36] are reflected in these differences, at least in some amino acids. In Supplementary Fig. 9, we present codon-specific Ramachandran plots for secondary substructures of the β-mode in an amino acid with large codon sample sizes (alanine). Substantial distribution differences are still observed between synonymous codons, even with such finer secondary structure conditioning. This indicates that distinct codon propensities for sub-structures of a β-mode cannot fully explain their distribution differences observed in the full β-mode.

**Distances between dihedral angle distributions of synonymous codons hint at a correlation to features of the translation process.** Our findings remain silent regarding the origin of the observed differences in synonymous codon backbone dihedral angle distributions; in particular, the causation direction cannot be established unambiguously. It is tempting to speculate, however, that the translation process plays an active role in the observed effect. To illustrate this speculation, we considered how two features of the translation machinery correlate to the calculated distances between backbone dihedral angle distributions of synonymous codon pairs. Firstly, we demonstrate that the difference in the codon-specific translation speed between a pair of synonymous codons appears to positively correlate to the distance between their dihedral angle distributions (Fig. 5, left). Although

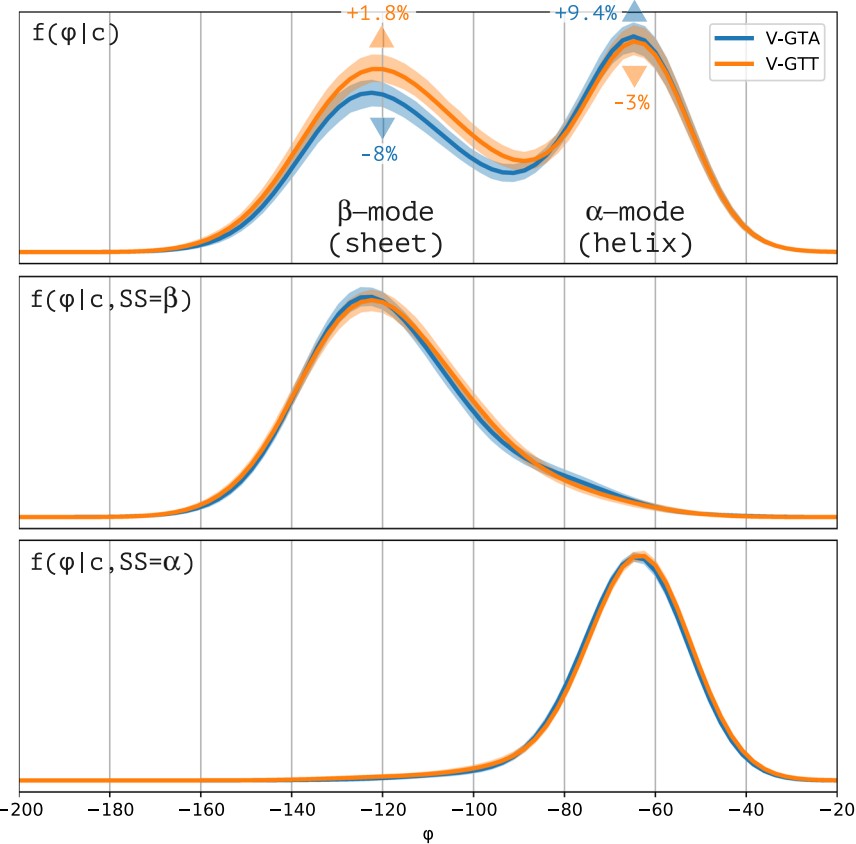

**Fig. 2 Different propensities for secondary structures of synonymous codons are manifested in the dihedral angle distribution.** Out of the two codons GTA and GTT translating valine, GTA has 8% lower propensity for the β mode and 9.4% higher propensity for the α mode. Propensities are manifested through the relative weights of the corresponding modes in the Ramachandran plot, which is visible in the marginal distributions of the dihedral angle φ plotted here. When conditioned by secondary structure (i.e., restricted to a specific mode), the distributions of the two synonymous codons become indistinguishable. By conditioning on secondary structure, our analysis is made robust to distribution differences arising from propensities differences. Kernel density estimates are shown with the shaded regions denoting 10–90% confidence intervals calculated on 1000 random bootstraps.

ribosome profiling has facilitated the measurement of translation speed to exquisite single-codon resolution in human and yeast cells, the application to bacteria has been more problematic[38]. We used the data from Chevance et al. who developed an in vivo bacterial genetic assay for measuring ribosomal speed independent of the stability of the mRNA transcript or the translated protein product[39]. Note that in order to limit confounding factors, we considered only pairs of codons being translated by the same tRNA.

In a second illustration, we identified codons translated unambiguously by a single tRNA, following Bjork et al.[40], and grouped codon pairs as being translated by either the same or different tRNA molecules. Figure 5 (right) shows that synonymous codon pairs translated by different tRNAs tend to have a larger distance between their backbone dihedral angle distributions.

While these two trends can by no means be conclusive, they suggest the potential value of the proposed methods in analyzing relations between synonymous coding and the features of the translation process.

## Discussion
In this work we generate codon-specific Ramachandran plots, showing that there is some association between synonymous codon usage and the structure of the translated amino acid. In contrast to previous works showing that synonymous codons

have preferences for different secondary structures, this work probes for a much more local association, namely between the backbone dihedral angle distributions of an amino acid and the synonymous codon from which it was translated. To factor out the phenomenon of secondary structure preference, we analyzed the α and β modes separately and found that many synonymous codon distributions are statistically significantly different in the β mode. We found no statistically significant differences in distributions for the α mode; perhaps not surprisingly given that α-helices fold into more rigid structures. Although our results cannot determine causal direction, it is worth clarifying that should synonymous codon usage be found to directly affect the formation of local protein structure this would not challenge the dominance of the amino acid sequence and protein environment in directing protein folding, especially for globular proteins having a well-defined fold. We would suspect that only some positions in a structure could carry a memory of potential structural bias introduced by synonymous coding, and that any environmental effects will not be biased towards any particular codon. This would mean that although the inability to factor out the environment is a limitation of our study, the differences between synonymous codon distributions would underestimate the effect that synonymous coding could have at positions, which are sensitive to this effect.

Given the mounting evidence for an association between codon usage and protein structure, it is not surprising that there have been previous attempts to combine genetic information with the

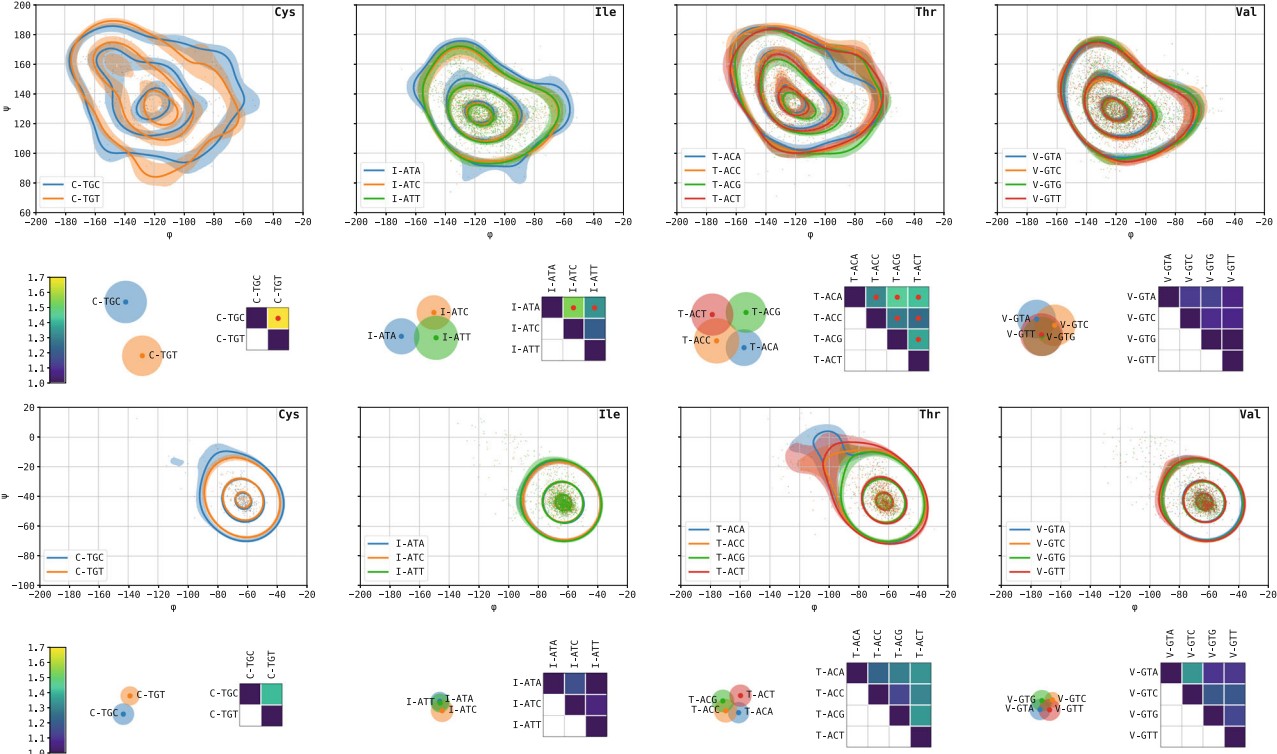

**Fig. 3 Codon-specific Ramachandran plots of select amino acids and distances between them.** Shown left-to-right are cysteine, isoleucine, threonine, and valine. Contour plots depict the level lines containing 10, 50, and 90% of the probability mass. Shaded regions represent 10%-90% confidence intervals calculated on 1000 random bootstraps. The β- (top) and α- (bottom) modes are depicted. The matrices show $L_1$ distances between pairs of codon-specific Ramachandran plots, normalized so that the self-distance is 1. Red dots indicate pairs with significantly different dihedral angle distributions based on their $p$ value. The scatter plots visualizing the distance matrices were obtained by a variant of multidimensional scaling (MDS). Each point represents a codon; pairwise Euclidean distances between the points approximates the $L_1$ distance between the corresponding codons. Circles approximate the uncertainty radii. The more two circles overlap, the less distinguishable are the corresponding codon-specific Ramachandran plots.

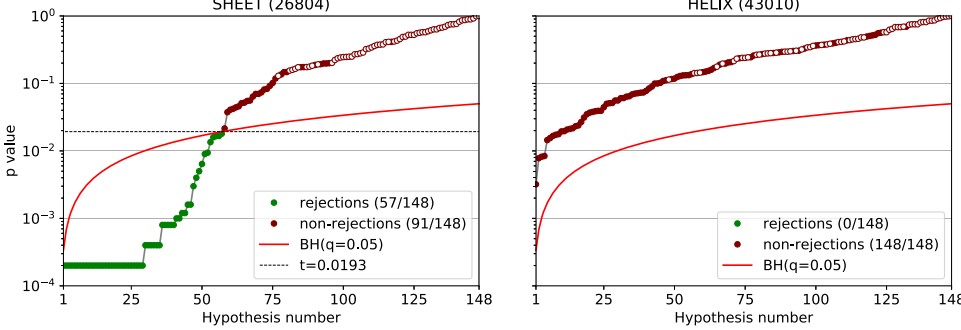

**Fig. 4 $p$ values obtained comparing pairs of synonymous codons in the β and α modes.** The $p$ values were obtained from the one-sided test detailed in the Methods. The total set of hypothesis tests included the 87 synonymous codon pairs with the addition of 61 comparisons of the codon with itself for control (denoted as empty circles). To correct for multiple tests, the rejection threshold corresponding to false discovery rate $q = 0.05$ was established using the Benjamini-Hochberg procedure (red curve). The set of tests on which the null was rejected is marked in green. For the identities of the rejected pairs, refer to Supplementary Fig. 4.

structural description of the protein products these genes produce. The Integrated Sequence-Structure Database (ISSD) catalogued in its second edition 88 *E. coli*, 25 yeast and 166 mammalian non-homologous proteins having a resolution better than 2.5Å[41] and the Cod-Conf Data Base assigned coding information to almost 1900 non-homologous proteins from all species[42]. Two important trends have eventuated in the two decades passed since the development of these databases: (1) there has been an exponential rise in the number of high-resolution protein structures, and (2) codon optimization has become common place in heterologous gene expression for

structural studies. This means that whilst we now have a wealth of structural data which *could* be used to explore associations with codons, they are not readily usable since structural databases, notably the PDB, rarely annotates the genetic sequence from which the protein was produced.

Codon-specific Ramachandran plots and their comparative analysis could serve as a useful, quantitative tool in future studies looking at the association between coding and local protein structure. It is likely that codon-specific backbone dihedral angle distributions will show even more significant variations when extended to pairs or triplets. Codon pair usage bias has been

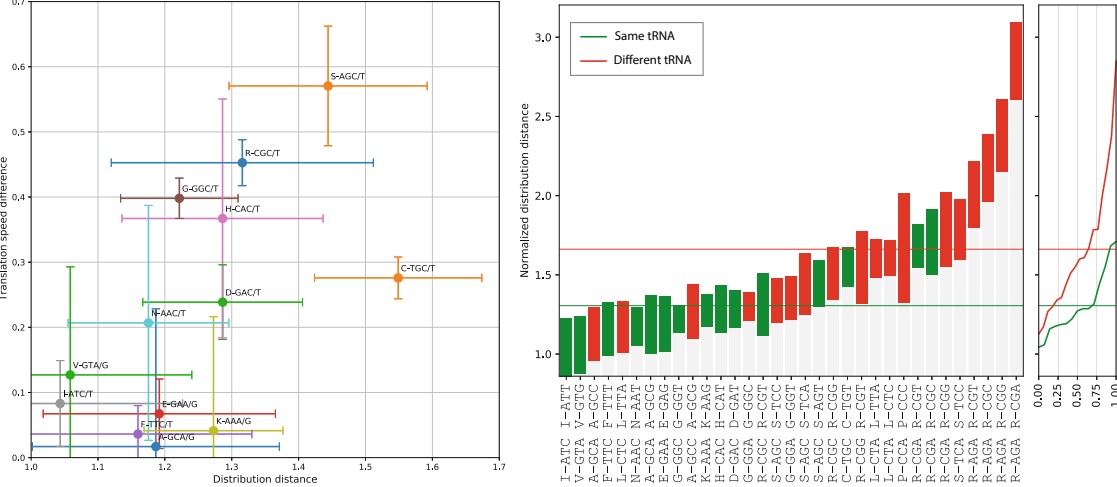

**Fig. 5 Distances between codon-specific Ramachandran plots are related to parameters of the translation process.** Left: The mean absolute difference in the relative translation speed as a function of the mean distance between backbone dihedral angle distributions for pairs of codons translated unambiguously by the same single tRNA. The two quantities are positively correlated ($r^2 = 0.6$). Translation speed data and confidence intervals are reproduced from Chevance et al. (2014). Translation speed (vertical) error bars were calculated from Fig. S1 A and B in Chevance et al. (2014) based on 3 or more independent assays; distribution distance (horizontal) error bars were obtained from 250 bootstrap samples. Both error bars indicate 1σ. Right: Pairwise distances between backbone dihedral angle distributions of codons translated unambiguously by the same tRNA (green) or two distinct tRNAs (red), sorted in ascending order (left) and as cumulative histograms (right). Noncognate codon pairs tend to exhibit a significantly bigger distance. Horizontal lines indicate means. In both plots, the normalized $L_1$ distances are reported with the ±σ confidence intervals calculated on 1000 bootstraps.

observed in *E. coli*[43] and in human disease[44,45]. It has been suggested that codon translation efficiency is modulated by adjacent single nucleotides[46], that codon pair order significantly affects translation speed[40], and, more recently, the case for a genetic code formed by codons triplets has been argued[24].

The main challenge in extending the presented methods to codon pairs or longer tuples is the relative scarceness of data and the need to compare multi-dimensional density functions characterizing the backbone structure of a tuple of amino acids. Extending the analysis to other expression systems faces a similar data scarceness challenge. Considering genes from various source organisms expressed in *E. coli*, either to probe for evolutionary distinctions between species or to overcome data scarceness when probing codon pairs in a hypothesized translation-dependent mechanism, is burdened by the uncertainty associated with codon (re)assignment. The latter will be overcome when structural databases start annotating the exact genetic source used for producing protein, which is crucial, given the ever-amounting evidence for the critical functional importance of codon usage.

We hope that the associations between synonymous coding and local backbone conformation revealed through codon-specific Ramachandran plots will spark subsequent investigations which should directly probe the possible causal relationships that might underpin these observations. The implications of an active, coding-dependent process would be tremendous, necessitating an immediate rethink as to how we manipulate the genetic code through codon optimization. This question could not be timelier, as mRNA vaccines are taking centre stage in global medicine. Moreover, the observed dependence between coding and local structure can potentially improve protein prediction algorithms, since in such tasks the causal relationship between the two is superfluous. To conclude, these results may affect how we define the role of synonymous variants in health and disease and understand protein folding in general.

## Methods

**Data collection**. Protein structure data is collected from the Protein Data Bank (PDB)[47] through a structured query against the search API defining the following criteria: (i) Method: X-Ray Diffraction; (ii) X-Ray Resolution: Less than or equal to

1.8 Å; (iii) $R_{free}$: Less than or equal to 0.24, (iv) Expression system contains the phrase "*Escherichia Coli*" and (v) Source organism taxonomy ID equal to 562 (*Escherichia Coli*). Queries return a list of PDB IDs having entity numbers, e.g., 1ABC:1. An entity corresponds to one or more identical polypeptide chains in the structure and we select the first of its matching chains using lexicographic order. A structure may have more than one unique entity (e.g., 1ABC:1 and 1ABC:2), in which case we would obtain both.

Next, we query the PDB's entry data API to obtain a mapping from chains to Uniprot[48] IDs. We keep only chains which map to a unique Uniprot ID, which is most chains. An exception, which we discard, are *chimeric chains* i.e. those that contain sections from multiple different proteins. We align the protein sequence of each chain to the Uniprot record sequence, using the same pairwise alignment algorithm as described in Codon assignment (below), to provide a Uniprot index for each residue in the PDB chain.

After removing homology bias using the procedure described under *Redundancy filtering*, backbone dihedral angles ($\varphi, \psi$) are calculated and secondary structure is assigned by DSSP[37] per residue. Finally, we assign each residue with a codon using the method described below (codon assignment). The result of this process is what we call a *Protein Record* for each PDB chain. The Protein Record contains, per residue: corresponding Uniprot ID and residue index, torsion angles ($\varphi, \psi$), secondary structure and codon.

*Redundancy filtering*. To remove homology bias from our data, we performed a filtering step. First, each pair of Uniprot sequences is aligned using the BioPython[49] software package, with a match score of 1 and all penalty scores set to zero. Thus, we obtain an alignment score $s_{ij} \geq 0$ between every pair of Uniprot sequences $i,j$. We then calculate normalized scores,

$$\widetilde{s}_{ij} = \frac{s_{ij}}{\sqrt{s_{ii}s_{jj}}}. \tag{1}$$

Note that by definition $0 \leq \widetilde{s}_{ij} \leq 1$ and $\widetilde{s}_{ii} = 1$ for every $i,j$. In other words, this normalization ensures that the self-alignment score is 1 and all other scores are normalized to be in [0,1], regardless of the sequence lengths or the alignment penalty values. This normalization also makes it simple to choose a similarity cutoff threshold, since the threshold is chosen in the fixed range [0,1] where 1 equates to an exact match and 0 to a complete mismatch. We chose a normalized similarity threshold of $\tau = 0.7$.

Using the normalized alignment scores we then employ a farthest-first traversal procedure to sort the Uniprot sequences: the first sequence is selected arbitrarily, and each successively selected sequence is such that it has the lowest maximum normalized alignment score between itself and all previously-selected sequences. Formally, denote by $\mathcal{S}$ and $\mathcal{U}$ the sets of selected and un-selected sequences, respectively. We initialize to $\mathcal{S} = \{0\}$ and $\mathcal{U} = \{1, 2, \ldots, N-1\}$ where $N$ is the number of Uniprot sequences. At each step $k$ of this traversal, for each unselected sequence $j \in \mathcal{U}$, we calculate its greatest similarity to any of the so-far selected

sequences,

$$S_k(j) = \max_{i \in \mathcal{S}} \widetilde{s}_{ij}. \tag{2}$$

We then choose the sequence which has the lowest maximal similarity to the selected sequences, i.e., we add

$$j = \arg\min_{j' \in \mathcal{U}} S_k(j') \tag{3}$$

to $\mathcal{S}$. We stop the procedure once $S_k(j) > \tau$ for the sequence $j$ that was selected at step $k$, and retain $\mathcal{S}$ as the output filtered set of Uniprot sequences. This ensures that no two sequences in the selected set have a normalized similarity score greater than $\tau$. After performing this procedure, we keep in our dataset only PDB chains that were mapped to one of the Uniprot sequences in $\mathcal{S}$. Any PDB chain mapped to a Uniprot sequence from $\mathcal{U}$ is discarded from analysis. Note that we keep all chains from different PDB structures that correspond to the same selected Uniprot sequence in order to aggregate their backbone angles as explained below (under *Angle aggregation*).

*Codon assignment.* Since the genetic sequences used for expressing each protein are not annotated in the PDB, we assigned codons from the native sequence in the European Nucleotide Archive (ENA)[50] IDs, cross-referenced from the mapped Uniprot ID. All available genetic sequences for the specific protein are translated to an amino-acid sequence and aligned pairwise to the sequence of the PDB chain. The alignment is performed using the BioPython[47] implementation of the Gotoh global alignment algorithm[51]. We used BLOSUM80 as the substitution matrix for the alignment, a gap-opening penalty of $-10$ and a gap extension penalty of $-0.5$.

Following the pairwise alignment of the amino acid sequence to all translated genetic sequences, we obtain the aligned codons from each sequence and assign them to corresponding residues from the PDB chain. This process yields zero or more assigned codons per residue in the PDB chain. In cases where more than one codon is assigned, we consider the assignment ambiguous and exclude that residue from further analysis.

*Angle aggregation.* Since some proteins have been characterized by multiple crystal structures, there are residues from different PDB chains which map to the same Uniprot ID and location in the Uniprot sequence. For example, in our dataset, the residues 1SEH:A:42, 1RNJ:A:42 and 2HRM:A:42 were all aligned to the Uniprot ID and index P06968:41. We consider such cases as different experimental realizations of the same protein residue and aggregate the backbone angles from such residues, to obtain an average measurement.

When aggregating the angles, we must account for the fact that a torsion angle pair $\boldsymbol{\varphi} = (\varphi, \psi)$ is defined on a torus (i.e. the domain $S^1 \times S^1$ where $S^1$ is a circle). Intuitively, each angle naturally wraps around at $\pm 180°$, and the space spanned by two such angles is a torus. Thus, taking a simple average of each angle separately would not be correct. Instead we use the torus-mean function defined in the mathematical tools section (below).

*Backbone angle distribution distance.* Our aim is to measure the distance between distributions of backbone torsion angles of synonymous codons in $\alpha$-helix and $\beta$-sheet secondary structure modes. Denote $f(\boldsymbol{\varphi}, \mathcal{X})$ the distribution of backbone angles $\boldsymbol{\varphi}$ of codon $c$ in secondary structure $\mathcal{X}$. We denote the distance between the backbone angle distributions of two synonymous codons $c$ and $c'$ in secondary structure $\mathcal{X}$ as $d(c, c')|\mathcal{X}$ and estimate them between all pairs of synonymous codons. We include all cases of $c = c'$ as controls. Empirical tests with the $L_1$, $L_2$ and smoothed Wasserstein distances showed that the $L_1$ metric provided the highest statistical power of all three at reasonable computational costs and so this was the distance metric selected. It is defined as,

$$d_1(c, c')|\mathcal{X} \triangleq \left\| f(\cdot|c, \mathcal{X}) - f(\cdot|c', \mathcal{X}) \right\|_1 = \int_{[-\pi, \pi]^2} \left| f(\boldsymbol{\varphi}|c, \mathcal{X}) - f(\boldsymbol{\varphi}|c', \mathcal{X}) \right| d\boldsymbol{\varphi}.$$

Although the underlying backbone angle distributions $f(\boldsymbol{\varphi}|c, \mathcal{X})$ are unknown, we sample from the distributions to obtain a finite sample $\{\boldsymbol{\varphi}_i \sim f(\cdot|c, \mathcal{X})\}_i$ for each codon $c$ and secondary structure $\mathcal{X}$. We use these samples to fit a kernel-density estimate (KDE), $\widehat{f}(\boldsymbol{\varphi}|c, \mathcal{X})$, of each distribution, as explained under *Kernel density estimation* (below). The distance metric $d_1(c, c')|\mathcal{X}$ is then calculated on the KDEs of each synonymous codon pair. Since the KDEs are discrete, the integration above becomes a sum,

$$\widehat{d}_1(c, c')|\mathcal{X} = \sum_{k_1, k_2 = 1}^{K} \left| \widehat{f}\left(\boldsymbol{\varphi}_{k_1, k_2}|c, \mathcal{X}\right) - \widehat{f}\left(\boldsymbol{\varphi}_{k_1, k_2}|c', \mathcal{X}\right) \right|,$$

where $K$ is the number of KDE bins in each direction and $\boldsymbol{\varphi}_{k_1, k_2} = \left(\varphi_{k_1}, \psi_{k_2}\right)$ are discrete evenly-sampled grid points. We then use permutation-based hypothesis testing to determine whether the distance supports the (alternative) hypothesis that the codons have a significantly different distribution, as explained below.

## Detecting synonymous codons with different angle distributions.
Faced with finite-sample estimations of codon backbone angle distributions, $\widehat{f}(\boldsymbol{\varphi}|c, \mathcal{X})$, we aim

to determine whether there exist pairs of synonymous codons $(c, c')$ for which the *underlying* distributions, $f(\boldsymbol{\varphi}|c, \mathcal{X})$, are different. For every pair of synonymous codons and secondary structure $(c, c')|\mathcal{X}$ (where we allow $c = c'$), we define a null hypothesis, which states that they have identical underlying backbone angle distributions:

$$H_{0,(c,c')|\mathcal{X}} : f(\boldsymbol{\varphi}|c, \mathcal{X}) = f(\boldsymbol{\varphi}|c', \mathcal{X})$$

We used permutation-based hypothesis testing[52] to obtain valid p-values for each of these null hypotheses without the need to make assumptions about the backbone angle distributions $f(\boldsymbol{\varphi}|c, \mathcal{X})$ or the distribution of the distance metric $d_1(c, c')|\mathcal{X}$ under the null. The permutation testing procedure is detailed below (under *Permutation-based two-sample hypothesis test*). We thus obtain, per secondary structure, a total of 148 p-values: 61 for identical codons, $c = c'$, and an additional 87 for non-idential but synonymous codons, $c \neq c'$.

We used the Benjamini-Hochberg method[53] for multiple hypothesis testing. In this approach a significance threshold is calculated dynamically from the set of all obtained *p*-values, in a way which controls the False-Discovery Rate (FDR) for the entire set of tests (instead of the type-I error of each individual test). The method allows us to specify the FDR-control parameter, $q$, and ensures that over repeated trials the expected value of the proportion between false discoveries (i.e. false rejections of the null hypotheses) and total discoveries (all rejections of null hypotheses) will be $q$.

*Preventing bias due to sample size differences.* One way to account for vastly different sample sizes this would be to cross-validate the KDE kernel bandwidth and choose an appropriate value for each sample size. This is challenging, however, since we would need to separately cross-validate for all codons, some with very limited data.

Instead, we opted to use a single kernel bandwidth, but fix the sample size for each set of synonymous comparisons. For each amino acid $\mathcal{A}$, and per secondary structure $\mathcal{X}$, we used the same, minimum, sample size $N_{\mathcal{A}, \mathcal{X}}$ to estimate the distributions for all codons in a synonymous group. Due to computational constraints, we also set an upper limit $N_{\max}$ of 200. Thus, the sample size for all codons of amino acid $\mathcal{A}$ was calculated as

$$N_{\mathcal{A}, \mathcal{X}} = \min\left\{ N_{\max}, \min_{c \in \mathcal{A}}\left\{ N_{c, \mathcal{X}} \right\} \right\},$$

where $N_{c, \mathcal{X}}$ is the sample size for codon $c$ in secondary structure $\mathcal{X}$.

Having limited the sample size, we employed also a bootstrapped-sampling[54] scheme on top of the distribution estimation and comparison, so as to exploit all available data for more common codons. Specifically, for each codon $c \in \mathcal{A}$, we estimate its distribution $B$ times from $N_{\mathcal{A}, \mathcal{X}}$ samples drawn with replacement from its collected data. This gives us access to at most $B \cdot N_{\mathcal{A}, \mathcal{X}}$ samples from each codon $c \in \mathcal{A}$, instead of only $N_{\mathcal{A}, \mathcal{X}}$. We then compare $B$ pairs of distributions for each synonymous codon pair $c, c' \in \mathcal{A}$ using the permutation test, and use the results of all permutations in all bootstrap iterations to calculate the p-value of $(c, c')$.

Statistical tests were performed with $B = 25$ bootstrap iterations with $K = 200$ permutations each, for a total of 5000 permutations used for $p$ value calculation. We used $N_{\max} = 200$ for all comparisons and set an FDR threshold of $q = 0.05$. Figure 6 presents a synthetic-data experiment validating this approach using various sample sizes.

*Full procedure.* The procedure for comparing synonymous codon backbone angle distributions and then identifying codon pairs having significantly different distributions is described here.

For each synonymous codon pair, $(c, c')$ and secondary structure $\mathcal{X}$, we calculate a *p*-value with respect to the null hypothesis $H_{0,(c,c')|\mathcal{X}}$, i.e. that they come from the same underlying distribution:
For $b \in \{1, \dots, B\}$:

Sample $N_{\mathcal{A}, \mathcal{X}}$ observations randomly from $c$ and from $c'$ (each with replacement).
Denote the sampled observations from $c$ and $c'$ as $\mathcal{C}$ and $\mathcal{C}'$ respectively.
Apply permutation test procedure (*Permutation-based two-sample hypothesis test*) on $\mathcal{C}$ and $\mathcal{C}'$ for $K$ permutations. The test-statistic $T(X, Y)$ first computes the KDEs of $X$ and $Y$, then calcultates the $L_1$ distance between them.
Denote by $\eta_b$ the number of times the base metric no greater than the permuted metric in the current permutation test.

Calculate the *p*-value with respect to $H_{0,(c,c')|\mathcal{X}}$:

$$p_{(c,c'),\mathcal{X}} = \frac{1 + \sum_{b=1}^{B} \eta_b}{1 + B \cdot K}.$$

For each secondary structure $\mathcal{X}$, we calculate the significance threshold based on the Benjamini-Hochberg method as follows:

Denote $\left\{ p_{i, \mathcal{X}} \right\}_{i=1}^{M}$ the set of $M = 148$ p-values obtained from all pairwise comparisons of synonymous codons in secondary structure $\mathcal{X}$.
Sort the p-values and denote $p_{(i), \mathcal{X}}$ the $i$-th sorted p-value.

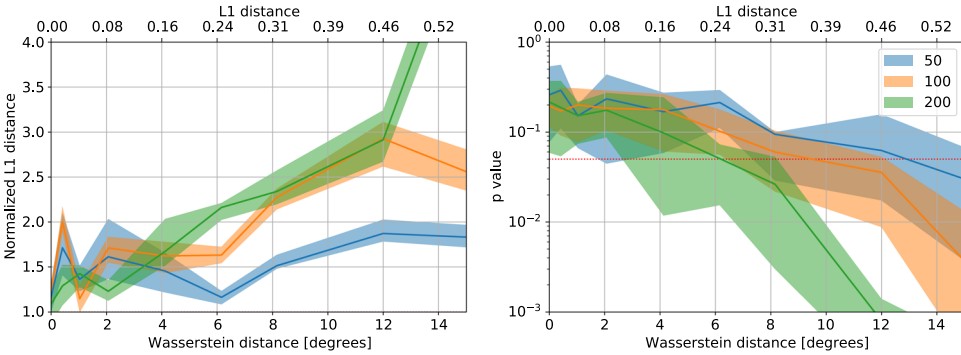

**Fig. 6 Normalized L₁ distance statistics and *p* values obtained comparing pairs of synthetic samples.** Samples were drawn from anisotropic von-Mises distributions with standard deviations of 35° in the φ direction and 18° in the ψ direction. One of the distributions was rotated by an increasing angle; the ground truth distance between the distributions was measured using the Wasserstein (W2) and L₁ metrics. Three sample sizes ($N = 50$, 100, and 200) are shown. Confidence intervals are 20%- and 80%-percentiles calculated on 10 random trials. Larger sample sizes allow to discern smaller distribution changes with higher significance.

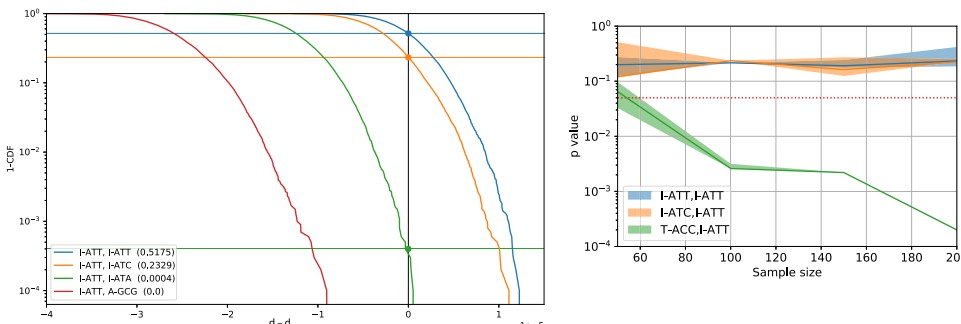

**Fig. 7 Example of test statistic distribution in the permutation test.** Left: Codon pairs are compared using the L₁ distance statistic between their dihedral angle KDEs in the β-sheet secondary structure mode. For each pair, depicted is one minus the cumulative distribution function (1-CDF) of the difference between the L₁ distance between the pair of KDEs and one between the pair of KDEs constructed with permuted labels. The intersection of 1-CDF with the vertical axis yields the p-value of the one-sided test (null hypothesis: d-d$_{perm}$ ≥ 0). When comparing a codon to itself (I-ATT, I-ATT), the null hypothesis holds, and the difference is expected to be positive half of the times (p-value≈0.5). The indistinguishable pair I-ATT, I-ATC produces a high *p*-value, while the more clearly distinguishable pair I-ATT, I-ATA yield a very low *p*-value. Two non-synonymous codons (I-ATT, A-GCG) appear perfectly distinguishable. Distributions were calculated using 100 bootstrap samples with 200 permutations in each. Right: *p*-values of pairwise comparison of a full sample of I-ATT in the β-mode (1365 samples) vs. different sample sizes of I-ATC and T-ACC ($N = 50$, 100, 150, and 200). The p-value of the distinguishable T-ACC, I-ATT pair decreases with the growth of the sample size. Confidence intervals are 20%- and 80%-percentiles calculated on 10 random trials.

Calculate the threshold p-value index for an FDR of $q$, which is the largest $p$ value smaller than the adaptive threshold of $q \cdot i/M$:

$$i_0 = \max\left\{ i : p_{(i), \mathcal{X}} \leq q \cdot \frac{i}{M} \right\}.$$

Set the adaptive significance threshold: $\alpha_M = p_{(i_0), \mathcal{X}}$.
Reject the *i*-th null-hypotheses if $p_{(i), \mathcal{X}} < \alpha_M$.

The set of synonymous codon pairs corresponding to the rejected null hypotheses are deemed to have significantly different backbone angle distributions. Figure 7 presents an experiment on real codon data, visualizing how the p-values calculated by our method correspond to expected differences between codons.

**Mathematical Tools**

*Torus mean.* Given a set of $N$ points on a torus $\{\boldsymbol{\varphi}_i\}_{i=1}^N$ where $\boldsymbol{\varphi}_i = (\varphi_i, \psi_i) \in S^1 \times S^1$, we would like to calculate the mean of these points, $\bar{\boldsymbol{\varphi}}$ in a way which accounts for the wrap-around of each angle at ±180°. We define a function which approximates a centroid on a torus, by calculating the average angle with circular wrapping in each direction separately. We denote this function as $\bar{\boldsymbol{\varphi}} = \text{torm}\left(\{\boldsymbol{\varphi}_i\}\right)$. For example, if $\boldsymbol{\varphi}_1 = (170, 170)$ and $\boldsymbol{\varphi}_2 = (-170, -130)$ then we expect $\text{torm}\left(\{\boldsymbol{\varphi}_1, \boldsymbol{\varphi}_2\}\right) = (180, -160)$. We define the function as follows

$$\bar{\boldsymbol{\varphi}} = \text{torm}\left(\{\boldsymbol{\varphi}_i\}_{i=1}^N\right) = (\bar{\varphi}, \bar{\psi})$$
$$= \left( \text{atan2}\left( \sum_{i=1}^N \sin\varphi_i, \sum_{i=1}^N \cos\varphi_i \right), \text{atan2}\left( \sum_{i=1}^N \sin\psi_i, \sum_{i=1}^N \cos\psi_i, \right) \right),$$

where $\text{atan2}(y, x)$ is a signed version of $\arctan(y/x)$ which uses the sign of both arguments to unambiguously recover the sign of the original angle $\theta$ such that $y = \sin\theta$ and $x = \cos\theta$.

*Torus distance.* Given two points on the torus, $\boldsymbol{\varphi}_1 = (\varphi_1, \psi_1)$ and $\boldsymbol{\varphi}_2 = (\varphi_2, \psi_2)$, we measure the distance, in angles between these points using the torus distance function as follows:

$$\text{tord}(\boldsymbol{\varphi}_1, \boldsymbol{\varphi}_2) = \sqrt{\arccos^2\cos(\varphi_1 - \varphi_2) + \arccos^2\cos(\psi_1 - \psi_2)}.$$

*Kernel density estimation.* We used two-dimensional kernel density estimation (KDE)[55] to estimate backbone angle distributions from finite samples. Given samples $\{\boldsymbol{\varphi}_i\}_{i=1}^N$ from torsion angles of a codon $c$ in secondary structure $\mathcal{X}$, we calculate

$$\widehat{f}(\boldsymbol{\varphi}|c, \mathcal{X}) = \frac{\gamma}{N} \sum_{i=1}^N K\left(\text{tord}(\boldsymbol{\varphi}, \boldsymbol{\varphi}_i)\right),$$

where $\boldsymbol{\varphi}$ represents points on a discrete grid, $K$ is a scalar kernel function, $\text{tord}(\cdot, \cdot)$ is the torus wrap-around distance defined under *Torus distance*, and $\gamma$ is a constant factor which normalizes the KDE so that it sums to one. The KDE was evaluated on a discrete grid of size 128 × 128, which corresponds to a bin width of $360/128 \approx 2.8°$. By applying the kernel to the wrap-around distance, we correctly account for the distance on the torus between each sample and each grid point. We used a simple univariate Gaussian kernel, $K(x) = \exp(-x^2/2\sigma^2)$, with a variance of $\sigma = 2$ (equivalent to the kernel bandwidth). We used a fixed bandwidth for all KDEs, ensuring to always compare KDEs calculated from the same number of samples.

*Permutation-based two-sample hypothesis test.* Given two statistical samples, $X = \{x_i\}_{i=1}^{N_X}$ and $Y = \{y_i\}_{i=1}^{N_Y}$ containing $N_X$ and $N_Y$ observations respectively, we wish to test whether the observations in both samples were obtained from the same underlying data distribution. A powerful and well-known approach to do this, is by conducting a two-sample statistical hypothesis test, with the null hypothesis that $X$

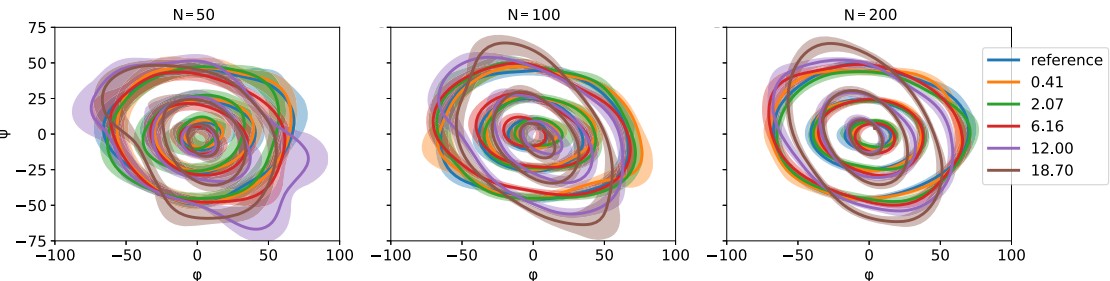

**Fig. 8 Ramachandran plots of synthetic distributions from Fig. 6.** Contours depict the level lines containing 10%, 50% and 90% of the probability mass. Shaded regions represent 10%-90% confidence intervals calculated on 1000 random bootstraps. The distributions are rotated one with respect to the other; the legend shows the ground truth Wasserstein ($W_2$) distance. Three sample sizes ($N = 50, 100$, and 200) are shown left-to-right. Larger samples lead to narrower confidence intervals.

and $Y$ are sampled from the same distribution, i.e., $H_0 : P_X(x) = P_Y(y)$. Such a test allows one to determine whether there is sufficient evidence to reject the null hypothesis, while limiting the chance of a type-I error (false positive, or rejecting $H_0$ when it is true) to be at most $0 < \alpha \ll 1$. Denote by $T(X, Y) \in \mathbb{R}$ a test statistic of our choosing, which numerically summarizes the differences between $X$ and $Y$, such that the smaller the value of $T(X, Y)$, the more $X$ and $Y$ are deemed similar. Further denote by $\hat{\tau} = T(X, Y)$ the value of this test statistic when evaluated on the samples at hand. To perform the hypothesis test, a p-value is calculated, which is the probability of obtaining a result at least as large as $\hat{\tau}$ under the assumption that $H_0$ is true: $p = \Pr[T \geq \hat{\tau}|H_0]$. The null hypothesis $H_0$ is then rejected if $p < \alpha$, thereby limiting the probability of type-I error to be $\alpha$.

In order to avoid making unfounded assumptions about the data or compromising on the choice of test-statistic, we employed a permutation-based two-sample hypothesis test[50], where the distribution of $T|H_0$ can be estimated for any choice of $T$ by randomly permuting the observations' labels. The procedure can be described as follows:

Inputs: samples $X = \{x_i\}_{i=1}^{N_X}$, $Y = \{y_i\}_{i=1}^{N_Y}$, test-statistic $T(X, Y) \in \mathbb{R}$, number of permutations $K$.
Compute the base statistic value: $\hat{\tau} = T(X, Y)$.
Pool the observations: $Z = \{z_1, \ldots, z_{N_X+N_Y}\} = \{x_1, \ldots, x_{N_X}, y_1, \ldots, y_{N_Y}\}$.
Compute a random permutation $\pi$ of $\{1, \ldots, N_X + N_Y\}$, such that $\pi(i)$ is the $i$-th element of this permutation.
For $k \in \{1, \ldots, K\}$:

Permute the pooled observations: $Z^\pi = \{z_{\pi(1)}, \ldots, z_{\pi(N_X+N_Y)}\}$.
Split the permuted observations:

$$X^\pi = \{z_{\pi(1)}, \ldots, z_{\pi(N_X)}\}$$
$$Y^\pi = \{z_{\pi(N_X+1)}, \ldots, z_{\pi(N_X+N_Y)}\}$$

Compute the permuted statistic value: $\tilde{\tau}_k = T(X^\pi, Y^\pi)$.
Calculate $\eta = \sum_{k=1}^{K} \mathbf{1}[\hat{\tau} \leq \tilde{\tau}_k]$, the number of times that the base statistical was no greater than the permuted statistic.
Calculate the p-value $p = \frac{1+\eta}{1+K}$.
Output: $p$ and $\eta$.

The key observation behind this approach is that under the null, we can treat $X$ and $Y$ as labels which are randomly assigned to observations from the same data distribution. Therefore, by permuting the labels and calculating the permuted test-statistic, we are obtaining samples of $T|H_0$. If $H_0$ is indeed true, we expect that $\hat{\tau} \approx \tilde{\tau}_k$, thereby yielding $p \approx 0.5$ as $K \to \infty$. Conversely, if $H_0$ is false, we would expect that $\hat{\tau} > \tilde{\tau}_k$, and then $p \to 0$ as $K \to \infty$. In practice, the number of permutations $K$ is limited by computational constraints. Nevertheless, since the smallest p-value which can be obtained is $p_{\min} = 1/(1 + K)$, we know the upper limit for the number of necessary permutations for a given significance level (in case of a single test).

**Pairwise distance plots**. For each secondary structure $\mathcal{X}$ and each amino acid $\mathcal{A}$, the statistical test procedure outlined under *Permutation-based two-sample hypothesis test* returns a matrix of all pairwise $L_1$ distance statistics $d(c, c')|\mathcal{X}$ averaged over bootstrap iterations, where $c, c'$ are two codons encoding $\mathcal{A}$. Since the used statistic is a metric, it is convenient to visualize it the form of a scatter plot in which each point represents a codon, and the Euclidean distances between each pair of points $c, c'$ approximate the distance statistic $d(c, c')|\mathcal{X}$. Such scatter plots are typically produced using multidimensional scaling (MDS)[56]. However, our case is different in the fact that the pairwise distance statistics are random variables, and the input data are finite sample approximations of their expected values. We

devised a variant of multidimensional scaling capable of handling this setting, which as far as we know is novel.

We aim at finding a collection of isotropic two-dimensional normal distributions $\mathcal{N}(\mu_c, \sigma_c^2 I)$ with locations $\mu_c$ and scales $\sigma_c$, each representing a codon $c$. The scales represent the uncertainty in location and constitute an extension of the standard MDS procedure which considers only locations. A simple calculation shows that the difference between two samples randomly drawn from $\mathcal{N}(\mu_c, \sigma_c^2 I)$ is itself normally distributed with zero mean and covariance $2\sigma_c^2 I$. Consequently, the squared Euclidean distance $d_2^2(c, c)$ is distributed as $2\sigma_c^2 \cdot \chi_2^2$, where $\chi_2^2$ denotes the chi-squared distribution with two degrees of freedom. The expected value of the latter squared distance is given by $4\sigma_c^2$ and should approximate the square of the measured statistic $d^2(c, c)|\mathcal{X}$ (the latter corresponds to the diagonal of the input distance matrix). We therefore determine the scale parameters by setting $\sigma_c = 0.5 \cdot d(c, c)|\mathcal{X}$.

A similar reasoning applies to the off-diagonal entries: the squared Euclidean distance $d_2^2(c, c')$ is distributed as $||\mu_c - \mu_{c'}||_2^2 + (\sigma_c^2 + \sigma_{c'}^2) \cdot \chi_2^2$, and its expectation is therefore given by

$$\mathbb{E}d_2^2(c, c') = ||\mu_c - \mu_{c'}||_2^2 + 2(\sigma_c^2 + \sigma_{c'}^2)$$

and should approximate $d^2(c, c')|\mathcal{X}$. Defining the target pairwise distances

$$\delta(c, c') = \sqrt{d^2(c, c')|\mathcal{X} - 0.5(d^2(c, c)|\mathcal{X} + d^2(c, c')|\mathcal{X})},$$

we now invoke a regular MDS to solve for the locations $\mu_c$.

Note that the procedure is exact when the input statistics are Euclidean distances between two-dimensional normal vectors; in other cases, the recovered locations and scales are merely an approximation of the underlying distributions.

For visualization completeness, we also report the averaged distance statistics. For convenience, the distances are normalized as $\frac{d(c,c')|\mathcal{X}}{\sqrt{d(c,c)|\mathcal{X} \cdot d(c',c')|\mathcal{X}}}$.

**Dihedral angle distribution plots**. The standard Ramachandran plot is often visualized either as a $\varphi, \psi$ scatter of the individual samples or as a density image estimated using KDE (the latter is sometimes plotted as level contours). Often, regions containing a certain amount of probability are superimposed. However, none of these visualization techniques represent the amount of uncertainty in the finite sample estimate of the probability density function. To capture the latter, we devised a new visualization (Fig. 8), described below.

Given a sample $\{\varphi_i\}_{i=1}^N$ of dihedral angles to visualize, we bootstrap $B$ independent samples of size $\min\{N, N_{\max}\}$. A normalized density image $f_b(\varphi)$ is constructed from each sample $b \in \{1, \ldots, B\}$ using the KDE procedure outlined under *Kernel density estimation*. The density images are averaged into a single density image $f(\varphi)$.

For the level contour $\lambda \in (0, 1)$, a threshold $\tau$ is calculated such that

$$\int_{\varphi \in [-\pi, \pi]^2 : f(\varphi) \geq \tau} f(\varphi) d\varphi = \lambda.$$

To calculate the uncertainty region of the above contour, we calculate the threshold $\tau_b$ for each density image $f_b(\varphi)$ individually and produce a set of binary images containing 1 wherever $f_b(\varphi) \geq \tau_b$ and 0 elsewhere; such images represent the $\lambda$-super level sets of the $f_b$'s. We then average these binary images and calculate their $\alpha$- and $(1 - \alpha)$-level sets. The region between these two contours is shaded in the plot and represents the $[\alpha, 1 - \alpha]$ confidence set.

In all our figures, unless specified otherwise, we used $B = 1000$ bootstraps with $N_{\max} = 200$; three levels $\lambda \in \{0.1, 0.5, 0.9\}$ were plotted with confidence set to $\alpha = 0.1$.

**Reporting Summary**. Further information on research design is available in the Nature Research Reporting Summary linked to this article.

## Data availability

The protein records data collected for this study as well as full output datasets have been deposited in the Harvard Dataverse database [https://doi.org/10.7910/DVN/5P81D4].

## Code availability

The code implementing the described data collection and analysis methods has been deposited in the Zenodo repository [https://doi.org/10.5281/zenodo.6345285]. Code is available under restricted access conditioned on user identification and agreement to academic use license.

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

## Acknowledgements

We are grateful to Joel Sussman and John Moult for their constructive skepticism and valuable comments. We thank Yaniv Romano for his helpful discussions on statistical methods.

## Author contributions

AM posed the original hypothesis; A.R., A.M. and A.B. designed the studies, interpreted the results and wrote the manuscript; A.R. and A.B. developed all the computational methods and performed the analyses. A.R. and A.M. contributed equally to this work.

## Competing interests

The authors declare no competing interests.
