## [Peer Review File · Nature Communications]

Codon-specific Ramachandran plots show amino acid backbone conformation depends on identity of the translated codonReviewers' Comments:

Reviewer #1:

Remarks to the Author:

Synonymous codon usage is now widely recognized as significant contributor to mRNA and protein homeostasis. It has been previously demonstrated that synonymous codons have non-random and distinct propensities to different secondary structures. However, investigating dependence between codon identity and the protein backbone structure has been previously hampered (in part) by the lack of sufficient structure and sequence information. In this manuscript, Rosenberg and colleagues compiled a database containing high resolution structures of the *Escherichia coli* proteins, the underlying amino acid sequences and, most importantly, nucleotide sequences of the encoded proteins. The final data set included 1343 protein chains. They have further developed a method for computing and comparing codon-specific Ramachandran plots and demonstrated that the backbone dihedral angle distributions of some synonymous codons are distinguishable with statistical significance for some secondary structures. These data revealed an important dependence between codon identity and the backbone torsion of the translated amino acid. The manuscript is well written and straightforward.

Comments:

1. Adzhubei and colleagues previously constructed a non-homologous database, termed the Integrated Sequence-Structure Database (ISSD), which comprised the coding sequences of genes, amino acid sequences of the corresponding proteins, their secondary structure and straight phi,psi angles assignments, and polypeptide backbone coordinates (Adzhubei et al, NAR, 1998 and 1999). The second edition of this database (ISSD2.0) had 88 non-homologous *E. coli* proteins, 25 yeast *Saccharomyces cerevisiae* proteins and 166 mammalian proteins. The authors should cite these previous attempts to construct a similar database and briefly comment on the differences between the databases, besides the obvious (i.e. number of proteins).
2. Most of the figures is difficult to read (even at 200% magnification), the authors should enlarge them (at least increase the font size).
3. The authors found the difference in synonymous codon propensity for the main two (alpha and beta) secondary structure types. However, it remains unclear whether there was any propensity dependence with respect to the position of the codons within the structure(s) (beginning/middle/end) reported previously (Adzhubei et al, FEBS Lett, 1996).
4. GTA is a relatively rare codon in *E. coli* and is expected to be slowly translated, while GTT is frequent and would be expected to be translated fast. The authors found that GTA has 8% lower propensity for strands and 9.4% higher propensity for helices. To what extent these preferences would affect the rates at which these two different secondary structures would be formed on the ribosome? The authors should comment on potential differences in the rates of the synthesis of the alpha-helices and beta structures based on their findings.
5. The discussion is a bit vague and very short and a better comparison with the previously reported observations showing non-random and distinct propensities of the synonymous codons to different secondary structures is required.
6. Stereochemical analysis of ribosomal transpeptidation performed previously by Lim and Spirin (J Mol Biol, 1986) allowed to suggest that the ribosome may generate an alpha-helical conformation at the C-end of the nascent peptide, as the only one unique conformation of the tetrahedral intermediate, found to be sterically compatible with all 400 possible pairs of the reacting amino acid residues and at the same time to be capable of cleaving into a planar trans-peptide group was found during this analysis and appeared to be similar to that in alpha-helix (given the torsion angles phi and psi). The authors should comment on this earlier observation in relation to their findings. This may also help them to develop a model explaining synonymous codon preferences to different secondary structure types.
7. It would be interesting to see the same analysis done for eukaryotic species and understand, whether the trend is just codon-type dependent, and/or the synonymous codon frequency is also a factor contributing to the choice. However, I clearly understand that this task is beyond the scope of

the present manuscript.

Reviewer #2:

Remarks to the Author:

The genetic code is degenerate; an amino acid can be translated from distinct codons. This study investigates the hypothesis that the backbone dihedral angle distribution of the 3D protein product is codon-specific. The authors analyzed high resolution crystal structures from the protein data bank. The study finds a statistically significant correspondence between codon identity and backbone torsion of the translated amino acid in beta sheets, but not in alpha-helices.

The work addresses an interesting question. While the effect of codons on the kinetics of translation is well-documented, their effect on local conformational variation has, to my knowledge, not been studied. The study's findings are intriguing and unexpected, at least to this reviewer. The methodology and analysis are thorough and described in great detail. A major weakness is that the data and findings are only sparsely documented. The manuscript would benefit from including more details, in particular describing the data:

- Please include a table with identities of all 1,343 protein chains
- Please include a table with the number of observations per codon for each residue.
- Please include a supplemental data set with the Protein Records for each chain.
- Fig 3 should also include a graphical representation of the pairwise distances (MDS?) as an illustrative example

Regarding statistical hypothesis testing. The analysis formulates (pairwise) null hypotheses within each amino acid class and within each secondary structure. The multiple pairwise hypotheses testing pertains to whether synonymous codons within an amino acid class lead to differentiated backbone angles. However, the authors designed the multiple hypotheses testing procedure to correct for all 148 hypotheses within a secondary structure. It seems to me they should have performed 18 different multiple hypotheses corrections: one for each amino acid across its synonymous codons (for each secondary structure). Please correct this.

'No synonymous pairs were rejected in comparisons of the distributions of the α -mode'. The stark contrast between alpha-helices and beta sheets is striking. I wonder if this can be in part attributed to co-translational folding of (local) secondary structure? Anti-parallel sheets, like alpha-helices, tend to be contiguous in sequence (and more stable), and therefore more directly affected by co-translational folding than parallel sheets. What do the results look like when they are performed separately on parallel and anti-parallel beta-sheets?

The codon-effect on backbone torsions appears small, but I would assume that rotamers will be affected as well. Have the authors considered that? The manuscript should include a discussion why the analysis focused on the phi/psi backbone dihedral angles, and not, for example on rotamers or include the peptide bond.

Several minor points:

Line 218: 'We consider such cases as different realizations of the same protein residue and aggregate the backbone angles from such residues, to obtain an "average" measurement of their backbone angles.' Should that be 'different realizations of the same codon'?

Line 228: 'there residues from different PDB chains which to the same Uniprot ID and location in the Uniprot sequence.' -> 'there are residues from different PDB chains which map to the same Uniprot ID and location in the Uniprot sequence.'

Line 266: the the

Line 308: that that

Line 310: at most q?

Line 297: indetial -> identical

Henry van den Bedem

Reviewer #3:

Remarks to the Author:

Rosenberg et al use a complex set of statistical analysis to seek possible correlations between the codon type used for an amino acid and the specific phi,psi angles it adopts when it is in a beta-sheet or when it is in an alpha-helix. They find no statistically significant differences in helical residues but see statistically significant differences for many codons in beta-sheet structures. I cannot evaluate the statistical methods, but something that concerns me is the clarity in the methods section of the measures they needed to take because of the sparseness of the data and not being able to assume anything about the underlying distributions. This means that (as far as I understand it) the analyses are based on many simulated data sets based on distributions that are inferred from sparse to very sparse data sets.

From my perspective as a protein structure expert, a number of things – all but one related to the sparseness of the data – lead me to think that these results are not meaningful but are simply the result of the sparseness of the data.

-The key point is that each residue in a protein structure is in a unique environment and its phi,psi angles are very strongly influenced by its environment. Then, since it is never the case in this non-redundant dataset that a given residue in a given environment is encoded by different codons, there is no ability to distinguish what differences in phi,psi angles are due to the environment and what are due to the codon. On top of this the sample sizes are small (up to only 200) and the phi,psi space in the beta region is large so that especially for the smaller populations there is no way that the distribution of residues encoded by a codon has enough sampling to be truly representative of all conformations that codon could adopt. For instance, looking at Figure 3 the second panel (For Ile) one can see that for the ATA (blue) codon there are literally just two observations near $\phi \sim -90 / +80$ and one observation near $\phi \sim -140 / =70$ that lead the 90%ile shape to look so different near $\psi \sim 80$. When just one or two observations that come from a unique protein structure wield such influence on the modeled distribution, that is a problem. It is believable that the distribution shows up as statistically different, but that statistical difference need not be related to the codon used but simply to the unique protein environments in which those few codon-encoded residues reside.

-notably the sparseness of the data is a much greater problem for beta-sheet residues compared with alpha-helical residues because the phi,psi region covered is so much larger. This is consistent with the authors possibly having enough data to provide robust “environmentally-averaged” populations of the amino acids from each codon in alpha-helices, but not in beta-sheets.

-A very strong indicator that sparseness of data is influencing these results is seen in the right-hand panel of Figure 4. The authors did not provide in the main manuscript the number of observations for the codon pairs but I did find them for each amino acid reported in supplemental Figure 1. These are crucially important information and so should be in a table somewhere - perhaps in the methods section. In ny case, quite striking and not noted by the authors is that ALL 16 of the codons to the

farthest right of the plot – i.e. those pairs with the greatest normalized phi,psi distances and including all with distances >1.5 – represent data-poor codons, having less than the ≥ 200 observations of the data-rich codons. These are the residues Arg (the most data-poor residue at just 27 observations), Ser (with 145 observations), Pro (61 observations), Leu (with 94 observations), and Cys (148 observations). Also noteworthy is that in the lefthand panel of Figure 4, the three codons with the highest distances and that play a big role in creating the observed correlation are the data-poor residues Cys, Ser and Arg.

-A fourth concern of mine, unrelated to the population sizes is also visible in the Figure 3 plots. This is that the variations seen in the distributions, even while statistically significant are small on an absolute scale, seemingly in most cases 5 degrees or less in phi and psi. That level of difference in a phi or psi angle is roughly matching the uncertainty of structures at near 1.8 Å resolution.

- If the authors want to pursue this relationship further, I note that the observation level for codon pairs was said to range over 100-fold from the most to the least observed codon pair (p. 17, line 315). Since Arg was the least observed codon pair at 27, I infer that means there is at least one codon pair that is observed 2700 times. My suggestion would be for the authors to carry out their analysis with the most populated codon pairs at varying observation levels, so 50, 100, 200, 500, 1000, 2000 and see how their results vary. If the level of significance of the distance variations shrinks as one goes to higher and higher numbers that would indicate that the low populations are the source of the signal rather than a true difference due to codons, and that could make an interesting contribution on its own. Furthermore aside from such statistical analyses, a simple (for a structural biologist) very direct experimental way to test the relative influence of the codon versus the protein environment on the phi,psi angles, would be to take any of these E. coli proteins, change one or more codons to synonymous ones, express the protein solve the structure and see the extent to which the phi,psi angles change.

Reviewer #4:

Remarks to the Author:

Authors show that phi and psi backbone torsion angles in proteins depend on which of synonymous codon encrypted particular amino acid. They were able to demonstrate this dependence in residues forming beta-sheets but not alpha-helices. Residues from loop regions were not considered in this study.

The connection between the type of codon used for the amino acid and backbone dihedral angles is a new observation. Authors state that there more connections could be found when the databases grow in size, connected and curated better.

Authors used sound methodology and described it in enough details to be reproducible. The evidence presented is enough to support the results claimed in the paper.

I would recommend publishing this article with minor corrections below:

Line 58: extra "comparing"

Fig 3: "b-strand" and "b-sheet" used for the same inconsistently

Fig S1, S2: Add type of residue, maybe with 3-letter codes

Fig S3, S4: Please change one-letter residue code in the caption to 3-letter.

Line 94-98: Numbers 57 and 87 do not seem to match the Fig S3. I counted total 84 synonymous pairs among which 28 were not rejected, 56 are rejected.

Figures 5-6 are not referenced in the text.

Response to Reviewers

We would like to thank all the reviewers for their detailed and constructive comments and interesting insights on our manuscript entitled "Codon-specific Ramachandran plots show amino acid backbone conformation depends on identity of the translated codon." (NCOMMS-21-45798). We have substantially reworked the manuscript and provided additional data as requested to address all the major concern raised by the reviewers. In what follows, we provide detailed answers to the specific questions and issues.

Reviewer #1

1. Adzhubei and colleagues previously constructed a non-homologous database, termed the Integrated Sequence-Structure Database (ISSD), which comprised the coding sequences of genes, amino acid sequences of the corresponding proteins, their secondary structure and straight phi,psi angles assignments, and polypeptide backbone coordinates (Adzhubei et al, NAR, 1998 and 1999). The second edition of this database (ISSD2.0) had 88 non-homologous E coli proteins, 25 yeast *Saccharomyces cerevisiae* proteins and 166 mammalian proteins. The authors should cite these previous attempts to construct a similar database and briefly comment on the differences between the databases, besides the obvious (i.e. number of proteins).

Unfortunately, it appears that the database, ISSD, is no longer available. However, we have now referenced in our expanded discussion this work, and another, as important earlier attempts to build databases including both coding and structural information side by side. Of course, the major technical difference between this database and the data we collect is the significantly increased number of structures and their resolution (2.5Å for ISSD vs 1.8Å in ours) and this is not surprising given the exponential increase in structural data over the last few decades. Beyond expansion of the amount and quality of data, the most important difference between these early works and the current work is the hypothesis being explored and the tools being used to do so. According to the articles you mentioned, the ISSD was designed and used to carry out direct analysis of the synonymous codon distribution frequencies between protein secondary structure types (and indeed for such a purpose lower resolution structures are useful). In contrast, we developed a data collection and analysis pipeline to probe for an association between coding and structure at a much more local level than secondary structure, namely the individual phi and psi of the encoded amino acid. One of the important steps in our data analysis is to condition on secondary structure to eliminate the recognized effect of secondary structure propensity and allow to probe for a more local and direct relationship.

2. Most of the figures is difficult to read (even at 200% magnification), the authors should enlarge them (at least increase the font size).

We have modified the figures for better legibility.

3. The authors found the difference in synonymous codon propensity for the main two (alpha and beta) secondary structure types. However, it remains unclear whether there was any propensity dependence with respect to the position of the codons within the structure(s) (beginning/middle/end) reported previously (Adzhubei et al, FEBS Lett, 1996).

We apologize for having missed this reference as one of the earlier works exploring codon preference in secondary structure, and it is now cited. As we mention in point 1, we condition on secondary structure to eliminate the part of the codon distribution difference which is due to secondary structure propensity, thus allowing us to probe for a direct relationship between coding and the conformation of the encoded amino acid. It is indeed possible that the differences we observe between synonymous codon dihedral angle distributions are originating from preferences to different positions within the secondary structures, or perhaps as Reviewer 2 suggests the specific types of secondary structure (parallel vs antiparallel or sheet vs strand). We mention this now as another direction for future exploration in the discussion but believe that carrying out such an analysis now is outside the scope of this work using the current data collection and analysis tools. Exploring these questions would require partitioning the (sometimes already small) samples into smaller samples which presents nontrivial challenges.

4. GTA is a relatively rare codon in *E. coli* and is expected to be slowly translated, while GTT is frequent and would be expected to be translated fast. The authors found that GTA has 8% lower propensity for strands and 9.4% higher propensity for helices. To what extent these preferences would affect the rates at which these two different secondary structures would be formed on the ribosome? The authors should comment on potential differences in the rates of the synthesis of the alpha-helices and beta structures based on their findings.

It has been reported previously that synonymous codons often have varied propensities for different secondary structures; this is not a new result. It has also been suggested previously that different secondary structures are preferentially coded by either translationally faster or slower mRNA regions (e.g., Thanaraj and Argos, 1996: Protein secondary structural types are differentially coded on messenger RNA). We do not include a discussion on this since it is outside the focus of the manuscript – as clarified above, we compared distributions after conditioning on the secondary structure to factor this known phenomenon out and allow for the detection of a more intrinsic and local association between codon identity and the backbone torsion of the encoded amino acid. The purpose of Figure 2 is to visualize the known phenomenon and to explain why we compare distributions for specific secondary structures individually.

5. The discussion is a bit vague and very short and a better comparison with the previously reported observations showing non-random and distinct propensities of the synonymous codons to different secondary structures is required.

Following the specific comments above and by other reviewers we have expanded our discussion section.

6. Stereochemical analysis of ribosomal transpeptidation performed previously by Lim and Spirin (*J Mol Biol*, 1986) allowed to suggest that the ribosome may generate an alpha-helical conformation at the C-end of the nascent peptide, as the only one unique conformation of the tetrahedral intermediate, found to be sterically compatible with all 400 possible pairs of the reacting amino acid residues and at the same time to be capable of cleaving into a planar trans-peptide group was found during this analysis and appeared to be similar to that in alpha-helix (given the torsion angles phi and psi). The authors should comment on this earlier observation in relation to their findings. This may also help them to develop a model explaining synonymous codon preferences to different secondary structure types.

One of the important aims of our research program is to mechanistically probe the effect of codon usage in structure. In this manuscript, we show only a statistical relationship for the association of coding and very local protein backbone structure. This evidence of an association cannot differentiate between the two possible causal directions: coding affecting

structure through the cellular translation machinery or structure affecting coding through evolution. In the final part of the manuscript, we show some association of our results to active features of the translation process, mostly to tempt others to join us in probing for such associations; the possibility that codons actively shape structure is very exciting and would have countless implications for biology, some of which we mention in the discussion. Beyond this, we believe it would be premature to direct any discussion towards potential mechanisms or models based on these results alone.

7. It would be interesting to see the same analysis done for eukaryotic species and understand, whether the trend is just codon-type dependent, and/or the synonymous codon frequency is also a factor contributing to the choice. However, I clearly understand that this task is beyond the scope of the present manuscript.

We completely agree that comparisons across species will be fruitful in providing deeper insight into the association between coding and structure. However, we did not include such an analysis since the PDB does not include an annotation of the exact genetic sequence used in expressing the protein being structurally characterized. Codon optimization is so commonplace that it makes it almost impossible to devise an accurate and automatic way to assign codons in a large data set containing heterologously expressed protein structures.

Reviewer #2:

1. The manuscript would benefit from including more details, in particular describing the data:

- Please include a table with identities of all 1,343 protein chains
- Please include a table with the number of observations per codon for each residue.
- Please include a supplemental data set with the Protein Records for each chain.
- Fig 3 should also include a graphical representation of the pairwise distances (MDS?) as an illustrative example

We have now included all these details. We added two supplementary tables, tabulating the data at the structure level, at the residue level. These files show all the features of the data including the PDB and Uniprot ID codes, resolution, assigned codon, phi and psi dihedral angles and more. A third table showing the number of observations per codon in each secondary structure was also added as Supplementary Material.

We had originally considered presenting our results using an MDS plot however found this representation less intuitive to biologists. We now gladly add this illustrative description of codon distribution distance alongside each pairwise matrix. We designed our own variant of an MDS procedure capable of visualizing not only the pairwise distances but also their uncertainties. We hope this new visualization technique will be appreciated by our readership.

2. Regarding statistical hypothesis testing. The analysis formulates (pairwise) null hypotheses within each amino acid class and within each secondary structure. The multiple pairwise hypotheses testing pertains to whether synonymous codons within an amino acid class lead to differentiated backbone angles. However, the authors designed the multiple hypotheses testing procedure to correct for all 148 hypotheses within a secondary structure. It seems to me they should have performed 18 different multiple hypotheses corrections: one for each amino acid across its synonymous codons (for each secondary structure). Please correct this.

Whilst the suggested amino acid specific multiple hypothesis testing can certainly be carried out as a complementary analysis, we think that designing the multiple hypothesis test to include all synonymous codon pairs simultaneously better addresses our null hypothesis which makes the general assertion that synonymous coding does not affect backbone angle distributions. In other words, by designing the test as we did, our aim is to reject the global null which states in our case that *any two* synonymous codons have the same underlying distribution. Moreover, this design is stricter, in the sense that we need to obtain lower p-values in order to get rejections, compared to the suggested approach. This is because the adaptive threshold becomes smaller as the number of hypotheses increases. We believe that showing numerous rejections under the stricter conditions is a stronger and more convincing result.

3. 'No synonymous pairs were rejected in comparisons of the distributions of the α -mode'. The stark contrast between alpha-helices and beta sheets is striking. I wonder if this can be in part attributed to co-translational folding of (local) secondary structure? Anti-parallel sheets, like alpha-helices, tend to be contiguous in sequence (and more stable), and therefore more directly affected by co-translational folding than parallel sheets. What do the results look like when they are performed separately on parallel and anti-parallel beta-sheets?

Alpha helices have a more rigid and defined structure than beta sheets, which should make this secondary structure less likely to carry any memory of subtle codon induced conformational preferences. This may explain why alpha helices showed less variance in the distributions. Indeed, it is likely that the greatest effect will be seen in the loop regions but here we need a way to account for the protein environment and as we discuss in the point 2 response to Reviewer 3, this is an active work in progress. It is possible that the differences we detect between synonymous codon distribution in the beta region are affected by codon preference for positions in the strand (as suggested by Reviewer 1) or as you suggest, by the type of sheet. Indeed, we mentioned this in the original manuscript "it is possible that some of the differences we observe between codon-specific dihedral angle distributions in the β -mode are attributable to codon preferences for finer secondary structure categories such as parallel and antiparallel β -sheets". This is certainly a direction we will pursue in the future, and we mention this now another direction for future exploration in the discussion. However, we believe that carrying out such an analysis now is outside the scope of this work using the current data collection and analysis tools. Exploring these questions would require both (1) a means for categorizing the beta region residues into different groups and this is a feature we do not have implemented in our current data collection and analysis pipeline and (2) partitioning the (sometimes already small) samples into smaller samples which presents nontrivial challenges.

Specifically, a note on the suggestion that some of our results might be attributable to co-translational folding of any sort. Any form of co-translational folding, for which there is much evidence, is the result of a codon exerting influence on the structure tens of residues upstream, at various positions which were previously translated and have already emerged from the ribosome exit tunnel. We find this unlikely to have a relation to the one-to-one association we used between codon and translated amino acid.

4. The codon-effect on backbone torsions appears small, but I would assume that rotamers will be affected as well. Have the authors considered that? The manuscript should include a discussion why the analysis focused on the phi/psi backbone dihedral angles, and not, for example on rotamers or include the peptide bond.

We do not challenge the dominance of the amino acid sequence or protein environment in folding a protein and rather find it likely that, especially in globular proteins having a distinct fold, any trace of the subtle effect of synonymous mutation will remain only at specific points, and also within the more restricted backbone chain. The side chains are much freer to rotate than the backbone and so will be more influenced by tertiary contacts, the formation of which would wash out any memory of codon induced structural bias.

Several minor points:

Line 218: 'We consider such cases as different realizations of the same protein residue and aggregate the backbone angles from such residues, to obtain an "average" measurement of their backbone angles.' Should that be 'different realizations of the same codon'?

This section described here is how we deal with the existence of several high-resolution structures for a particular protein. We aggregate the backbone angles for each residue across the multiple structures using a torus-centroid. Since the structures described here are all of the same protein from the same species (i.e., belong to a single and unique Uniprot ID) then indeed each residue is mapped to a single and unique codon as described in the methods section. As such we think the text is correct as written and have added the word "experimental" (i.e., "different experimental realizations") in case that makes the sentence clearer.

Line 228: 'there residues from different PDB chains which to the same Uniprot ID and location in the Uniprot sequence.' -> 'there are residues from different PDB chains which map to the same Uniprot ID and location in the Uniprot sequence.'

Line 266: the the

Line 308: that that

Line 310: at most q?

Line 297: indetial -> identical

Each of these typos have been corrected in the amended text.

Reviewer #3:

1. Something that concerns me is the clarity in the methods section of the measures they needed to take because of the sparseness of the data and not being able to assume anything about the underlying distributions. This means that (as far as I understand it) the analyses are based on many simulated data sets based on distributions that are inferred from sparse to very sparse data sets.

We improved the description regarding how we deal with the challenge of variable and sometimes small sample sizes. Moreover, to conclusively address the issue of sparse data, we conducted new experiments on synthetic data which demonstrate how our analysis methods work and showcase their sensitivity across multiple sample sizes in the ranges used in our analysis. Based on these experiments, we conclude that the data sets are big enough to allow accurate non-parametric comparison of the two-dimensional estimated distributions.

We would like to clarify our reasoning for not assuming anything about the distributions; it is an important advantage of our methodology and not a circumstance of the data at hand. We did not assume any specific parametric form of the underlying (i.e., real) distribution of codon dihedral angles because these distributions are complex, unknown, and unlikely to be accurately approximated by any closed-form parametric model. Thus, rather than relying on assumptions, we developed tools and employed existing statistical methods which are sensitive and capable of comparing between non-parametric estimated distributions.

Since no assumption is made on the data distribution, we also have to make no assumption about the distribution of our test statistics under the null hypothesis. We rather devise a statistical test based on permutations that is free of such assumptions, which is, in our view, another major advantage of the proposed methodology.

2. The key point is that each residue in a protein structure is in a unique environment and its phi,psi angles are very strongly influenced by its environment. Then, since it is never the case in this non-redundant dataset that a given residue in a given environment is encoded by different codons, there is no ability to distinguish what differences in phi,psi angles are due to the environment and what are due to the codon. On top of this the sample sizes are small (up to only 200) and the phi,psi space in the beta region is large so that especially for the smaller populations there is no way that the distribution of residues encoded by a codon has enough sampling to be truly representative of all conformations that codon could adopt. For instance, looking at Figure 3 the second panel (For Ile) one can see that for the ATA (blue) codon there are literally just two observations near $\phi \sim -90/+80$ and one observation near $\phi \sim -140/\sim 70$ that lead the 90%ile shape to look so different near $\psi \sim 80$. When just one or two observations that come from a unique protein structure wield such influence on the modeled distribution, that is a problem. It is believable that the distribution shows up as statistically different, but that statistical difference need not be related to the codon used but simply to the unique protein environments in which those few codon-encoded residues reside.

Firstly, we agree that a limitation of this type of analysis is that it fails to factor out the influence of the environment. To address this limitation, we are currently conducting another study where we automatically identify environmentally matched positions in different (including homologous) structures (short sequences of identical amino acids having similar tertiary interactions with the protein environment) and are quantifying the correspondence between variations in the backbone torsion across these short sequences and the coding of these sequences. Our preliminary results strongly support the association of coding with environmentally matched local backbone structure and once we have rigorously completed this work, which we believe is outside of the scope of the current manuscript, we will publish the findings separately.

Secondly, concerning the effect of environment on the current study. We do not challenge the dominance of the amino acid sequence or protein environment in folding a protein and rather find it likely that, especially in globular proteins having a distinct fold, any trace of the subtle effect of synonymous mutation will remain only at specific points. The crucial point for our analysis, however, is that the influence of the environment should not be associated with any particular codon. This means that any environmental influences will decrease, and not increase, any signal we have for differences between synonymous codon distributions. This is because they would essentially be adding “noise” to our angle measurements which is uncorrelated with the codon identity. As such, we estimate that in an environment-controlled setting, we would expect our results to show even more significant differences between synonymous codon distributions.

Thirdly, regarding the size of the beta region ostensibly preventing us from observing all possible codon conformations at the lower sample sizes. Indeed, it is the case that with smaller sample sizes, the ability to estimate the true underlying distribution diminishes. However, our task here is not to discover the true distributions, but only to compare them in the statistical-hypothesis sense: we assume two distributions *are the same* and calculate the p-value representing the probability of observing such samples from them under this assumption. To this end, we do not need the given samples to fully represent each possible conformation that exists in the true underlying distribution. To demonstrate this point further, and show that our statistical analysis holds for the sample sizes considered here, please refer to the new synthetic data experiments which have been added to the manuscript.

3. notably the sparseness of the data is a much greater problem for beta-sheet residues compared with alpha-helical residues because the phi,psi region covered is so much larger. This is consistent with the authors possibly having enough data to provide robust “environmentally-averaged” populations of the amino acids from each codon in alpha-helices, but not in beta-sheets.

Please see the previous answer regarding the size of the beta region and its effect on our statistical results.

Here we would like to point out that in the synthetic data experiments, we experiment with (among other things) the variance of the data distribution, i.e. we show what happens by taking the same distribution and spreading it over a larger space. The figure reported below is a version of the new synthetic experiment Figure 6, in which we compared the p-values for different sample sizes. The dotted lines show the p-values on the same data scaled down by a factor of 2 (approximately matching the spread of the alpha mode).

The results of this experiment show that spread does not prevent us from detecting the differences between two distributions and quantifying these differences as p-values.

Since alpha helices have a more rigid and defined structure than beta sheets, it would make this secondary structure less likely to carry any memory of subtle codon induced conformational preferences. Therefore, it is not surprising that alpha helices showed less variance in their distributions.

4. A very strong indicator that sparseness of data is influencing these results is seen in the

right-hand panel of Figure 4. The authors did not provide in the main manuscript the number of observations for the codon pairs but I did find them for each amino acid reported in supplemental Figure 1. These are crucially important information and so should be in a table somewhere - perhaps in the methods section. In any case, quite striking and not noted by the authors is that ALL 16 of the codons to the farthest right of the plot – i.e. those pairs with the greatest normalized phi,psi distances and including all with distances >1.5 – represent data-poor codons, having less than the ≥ 200 observations of the data-rich codons. These are the residues Arg (the most data-poor residue at just 27 observations), Ser (with 145 observations), Pro (61 observations), Leu (with 94 observations), and Cys (148 observations). Also noteworthy is that in the lefthand panel of Figure 4, the three codons with the highest distances and that play a big role in creating the observed correlation are the data-poor residues Cys, Ser and Arg.

We have now added a table to the Supplementary Materials which shows the sample size for each codon in each of the secondary structures.

With relation to Figure 4 (now 5), we would like to clarify that this is not one of the main results of our study but rather an element we chose to include, to suggest the potential association of our findings to previously described codon dependent translational features. This analysis was carried out in a way which highlights one of the features of the translation machinery, whilst limiting confounding factors. This means that each analysis considers only a subset of all possible codon pairs; either those having a codon pair translated by the same tRNA or those translated unambiguously by a single tRNA.

In the lefthand panel of Figure 4 (now 5) the codon pairs with the greatest distance distributions are the two most common codons of Arg (CGC and CGT with sample sizes of 490 and 448 respectively), the most common and fourth most common codons of Ser (AGC and AGT with sample sizes of 313 and 186 respectively) and the only two codons of Cys (having sample sizes of 191 and 148). It is certainly not the most data sparse codons which dominate this trend.

In the right hand panel we agree that this result represents some overlap of the following trends; increased codon pair distance for codons translated by different tRNAs and increased codon pair distance where one of the codons is rare. Since the purpose of this figure is to encourage others to consider our findings in connection to studies on translation mechanisms and given that we have shown that our analysis methods are not biased by sample size in the additional experiments described here and included in the revised manuscript we believe that it is fair to present this observation.

Concerning the issue of "data-poor residues". These numbers (e.g., 27 for Arg) correspond to the rarest codon in each of these amino acids, while other codons of these amino acids have larger sample sizes. When comparing distributions of synonymous codons, we indeed always use the sample size of the smallest codon per AA, to estimate the distributions from their samples. This prevents the distribution estimation itself from being biased. However, we also employ a bootstrap-resampling scheme (again, always resampling the number of samples equal to the smallest codon), which enables us to exhaust the samples of the larger groups and compare distributions estimated from them to the other codon. We do multiple comparisons for each re-sampling, and in this way, we can calculate whether the distributions are likely to be the same. As explained above, our statistical method assumes that every pair of synonymous codon distributions is identical and calculates the probability to see the data at hand under that assumption. This can be calculated even when one of the codons has a small sample size, as long as the samples are "different enough" from the other codons'. Whether they are "different enough" to reject the assumption is quantified by the p-value. The

synthetic data experiments we added in this revision demonstrate this point further. This is of course also explained in detail the methods.

5. A fourth concern of mine, unrelated to the population sizes is also visible in the Figure 3 plots. This is that the variations seen in the distributions, even while statistically significant are small on an absolute scale, seemingly in most cases 5 degrees or less in phi and psi. That level of difference in a phi or psi angle is roughly matching the uncertainty of structures at near 1.8 Å resolution.

The choice of resolution was of course a tradeoff between keeping sample sizes as large as possible on one hand and having accurate phi/psi angles on the other. We added a histogram showing the resolutions of all the structures used in our analysis. Since 1.8Å is our upper bound on the resolution, the median resolution is actually 1.6Å and the amount of experimental measurement noise is probably lower than 5 degrees. This is further supported by the fact that the median angle disagreement across multiple PDB structures of the same proteins is below 2.8 degrees (we added a corresponding histogram in Supplementary Figure 1).

Secondly and more importantly, the measurement in unrelated locations across different proteins is expected to be independent; the effect of such noise would amount to a “blur” in the dihedral angle distributions (technically, convolution of the angle density function with the noise density) which should have negligible impact on the calculated statistics. In fact, the amount of blur introduced by resampling and kernel density estimation is bigger and is necessary for obtaining reliable statistics from finite samples.

6. If the authors want to pursue this relationship further, I note that the observation level for codon pairs was said to range over 100-fold from the most to the least observed codon pair (p. 17, line 315). Since Arg was the least observed codon pair at 27, I infer that means there is at least one codon pair that is observed 2700 times. My suggestion would be for the authors to carry out their analysis with the most populated codon pairs at varying observation levels, so 50, 100, 200, 500, 1000, 2000 and see how their results vary. If the level of significance of the distance variations shrinks as one goes to higher and higher numbers that would indicate that the low populations are the source of the signal rather than a true difference due to codons, and that could make an interesting contribution on its own.

We performed the suggested experiment by taking the ATT codon in Isoleucine (after filtering the beta mode, 1365 samples remained) as the reference and comparing it to three other codons (I-ATT itself, an indistinguishable I-ATC, and a distinct ACC codon of another amino acid). The second sample being compared was randomly subsampled down to 50, 100, 150 and 200 points over 10 independent trials. The smaller subsample was compared to the big I-ATT sample using the same statistical testing procedure outlined in the paper.

Figure 7 (right) in the revised manuscript shows the obtained p values. As expected, the p value decreases (significance improves) for the distinguishable pairs of codons (ATT vs ACC) – in fact, 100 samples are abundantly sufficient to obtain a very reliable null rejection. On the other hand, for indistinguishable distributions (ATT vs ATT) and (ATT vs ATC), the p value stays high regardless of the sample size increase.

We think this experiment as well as the synthetic data experiment we added to the methods section should dispel the very important concern raised by the reviewer.

7. Furthermore aside from such statistical analyses, a simple (for a structural biologist) very direct experimental way to test the relative influence of the codon versus the

protein environment on the phi,psi angles, would be to take any of these E. coli proteins, change one or more codons to synonymous ones, express the protein solve the structure and see the extent to which the phi,psi angles change.

Indeed, we have recently established an experimental setup and are conducting such experiments. We note that it is not so simple as synonymously mutating any protein at any position and assessing the effect on phi-psi at the mutated places: we propose that, especially in well folded proteins having a single, global minima (exactly those proteins which crystallize to produce structures with high resolution and can therefore be found in the PDB), there will be only specific, relatively rare, points which will maintain the structural imprint of synonymous mutation.

Since the present analysis only makes claims about the *distributions* of synonymous codon dihedral angles, it is not constructive as to which are the *specific* proteins and locations to mutate. On the other hand, the complementary analysis we described in response to point 2 above, assessing the effect of coding on environmentally paired protein positions, will help guide our choice for proteins and positions to explore the effect of coding on structure.

Reviewer #4

1. I would recommend publishing this article with minor corrections below:

Line 58: extra "comparing"

Fig 3: "b-strand" and "b-sheet" used for the same inconsistently

Fig S1, S2: Add type of residue, maybe with 3-letter codes

Fig S3, S4: Please change one-letter residue code in the caption to 3-letter.

Figures 5-6 are not referenced in the text.

All of these corrections have been made in the amended text.

Line 94-98: Numbers 57 and 87 do not seem to match the Fig S3. I counted total 84 synonymous pairs among which 28 were not rejected, 56 are rejected.

We double checked this and the numbers in the text are correct. Note that rather than counting the points in Supplementary Fig. 4 (originally 3), it is possible to see the summary of rejections Fig. 4 (Originally 6).

Reviewers' Comments:

Reviewer #1:

Remarks to the Author:

The manuscript by Rosenberg and co-authors has been revised. Additional information has been added, which was missing in the original version of the manuscript. In sum, I feel that the authors have responded to the majority of the previous concerns either through additional data or changes to the text.

However, I strongly believe that a very important point as it relates to the propensity dependence with respect to the position of the codons within the structure(s) (beginning/middle/end) should be addressed by the authors in this manuscript (at least for the beta structures). I believe that this is an important point that may add substantial value to the current study given also recent findings by Agirrezabala et al, EMBO J 2022 showing that a switch from alpha-helical to beta-strand conformation is possible during co-translational protein folding and is occurring with the space of ribosomal tunnel.

Minor: Reference Adzhubei et al, FEBS Lett, 1996 appeared now to be duplicated (as Ref 30 and Ref 38).

Reviewer #2:

Remarks to the Author:

The authors have satisfactorily addressed all my comments

Reviewer #3:

Remarks to the Author:

The revised manuscript has been improved with a couple of test analyses with simulated and real data that now reported in Figures 6 and 7 (are the pointers from the text to Figs. 6 and 7 swapped?). They appear to show that the statistically significant correlation is not simply a result of small populations, although given that the small data sets are the ones that stand out the most, I am still wondering about that. I still think two things are important to very explicitly and clearly emphasize.

The first is to be very clear that the only thing that can be firmly concluded based on these analyses is that there is a "correlation" and NOT that there is a "cause/effect relationship" or a "dependence" of one on the other. The authors can speculate about that if they like, but it must be made much more clear that this is speculation. Right now, as far as I could see, the text does not explicitly make this point and in many places implies that there is a "dependence" between the two. For instance: from the abstract "This shows that there exists a dependence between codon identity and backbone torsion of the translated amino acid. Although these findings cannot pinpoint the causal direction of this dependence ..."; lines 44-45 state "a dependence between the codon identity and the backbone dihedral angle"; lines 156-7 "some association between synonymous codon usage and the structure"; lines 168-9 "only some positions in a structure will carry a memory of structural bias introduced by synonymous coding"; lines 171-2 "underestimate the effect that synonymous coding could have at positions which are sensitive to this effect".

The second point is to be clear that the differences in conformation are small, just a few degrees in phi/psi angles and so even if there is a cause/effect relationship, it is not clear how meaningful the differences in structures are in terms of their impact on protein folding and structure.

Response to Reviewers

We thank the reviewers for taking the time to carefully reassess our revised manuscript entitled "Codon-specific Ramachandran plots show amino acid backbone conformation depends on identity of the translated codon." (NCOMMS-21-45798) and were pleased to see that our revisions addressed almost all of the original concerns and comments. We have now also addressed the final remaining issues as detailed below.

Reviewer #1

I strongly believe that a very important point as it relates to the propensity dependence with respect to the position of the codons within the structure(s) (beginning/middle/end) should be addressed by the authors in this manuscript (at least for the beta structures). I believe that this is an important point that may add substantial value to the current study given also recent findings by Agirrezabala et al, EMBO J 2022 showing that a switch from alpha-helical to beta-strand conformation is possible during co-translational protein folding and is occurring with the space of ribosomal tunnel.

We have performed this analysis now and included the results in two additional Supplementary Figures, 8 and 9. Briefly, we observe the expected codon preference for specific positions in the beta sheet but also show for one amino acid having large synonymous codon samples that such preferences do not fully account for the differences we observe in Codon-specific Ramachandran Plots since differences remain in these plots even using this finer analysis. The sample sizes would be prohibitively small in many cases to carry out this analysis for all amino acids.

Minor: Reference Adzhubei et al, FEBS Lett, 1996 appeared now to be duplicated (as Ref 30 and Ref 38).

We have removed this duplication in the newly revised version.

Reviewer #3

The revised manuscript has been improved with a couple of test analyses with simulated and real data that now reported in Figures 6 and 7 (are the pointers from the text to Figs. 6 and 7 swapped?).

Indeed, they were mistakenly swapped. This has been fixed in the revised manuscript.

They appear to show that the statistically significant correlation is not simply a result of small populations, although given that the small data sets are the ones that stand out the most, I am still wondering about that. I still think two things are important to very explicitly and clearly emphasize. The first is to be very clear that the only thing that can be firmly concluded based on these analyses is that there is a "correlation" and NOT that there is a "cause/effect relationship" or a "dependence" of one on the other. The authors can speculate about that if they like, but it must be made much more clear that this is speculation. Right now, as far as I could see, the text does not explicitly make this point and in many places implies that there is a "dependence" between the two. For instance: from the abstract "This shows that there exists a dependence between codon identity and backbone torsion of the translated amino acid. Although these findings cannot pinpoint the causal direction of this dependence ..."; lines 44-45 state "a dependence between the codon identity and the

backbone dihedral angle”; lines 156-7 “some association between synonymous codon usage and the structure”; lines 168-9 “only some positions in a structure will carry a memory of structural bias introduced by synonymous coding”; lines 171-2 “underestimate the effect that synonymous coding could have at positions which are sensitive to this effect”.

We would like to point out that statistical dependence is a technical term which does not imply any causality. Two variables X and Y (e.g. codon identity and dihedral angle) are **independent** if (informally) knowledge of either one does not affect the probability distribution of the other. They are **dependent** when this property does not hold, i.e. knowledge of either changes the probability distribution of the other. Technically, when X and Y are independent it implies that both $P(Y|X)=P(Y)$ and $P(X|Y)=P(X)$, and therefore when they are dependent then either $P(Y|X)\neq P(Y)$ or $P(X|Y)\neq P(X)$, or both. Thus, causality cannot be determined from statistical dependence, and none is implied.

In our specific case, the use of “dependence” means that given the codon identity (e.g. X), the angle distribution (e.g. $P(Y)$) changes, i.e. $P(Y|X)$ is different from $P(Y)$, which is what we observed. Use of the term “dependence” rather than “correlation” here is more accurate because correlation is just one specific way to measure statistical dependence between variables (specifically, linear dependence), which was not an applicable method in this case.

We believe we were explicit in both the abstract (“...these findings cannot pinpoint the causal direction of this dependence...”) and discussion (“We hope... will spark new investigations which should directly probe the possible causal relationships that might underpin these observations...”) in stating that we cannot determine the direction of causality from these results. Still, to avoid any potential confusion we now (1) clarify in the first description of main result that the statistical dependence cannot determine causality and (2) have amended the end of the first paragraph in the discussion to first clearly state that we cannot say anything about causality based on these results before describing the potential biological significance should it be found that coding affects structure. We believe these additional clarifications now accurately and fairly describe our results.

The second point is to be clear that the differences in conformation are small, just a few degrees in ϕ/ψ angles and so even if there is a cause/effect relationship, it is not clear how meaningful the differences in structures are in terms of their impact on protein folding and structure.

Understanding the implications of these results for protein folding, structure and function will require dedicated studies and we call for such studies in conclusion. We also note in the discussion that should it be found that coding does affect structure directly then it will probably only occur at specific points and so the small effects observed at the distribution level could be much more pronounced at specific positions.